# Digital cell quantification identifies global immune cell dynamics during influenza infection

Zeev Altboum[1,†], Yael Steuerman[2,†], Eyal David[2], Zohar Barnett-Itzhaki[1], Liran Valadarsky[1], Hadas Keren-Shaul[1], Tal Meningher[3,4], Ella Mendelson[3,5], Michal Mandelboim[3,*,§], Irit Gat-Viks[2,**,§] & Ido Amit[1,***,§]

## Abstract

Hundreds of immune cell types work in coordination to maintain tissue homeostasis. Upon infection, dramatic changes occur with the localization, migration, and proliferation of the immune cells to first alert the body of the danger, confine it to limit spreading, and finally extinguish the threat and bring the tissue back to homeostasis. Since current technologies can follow the dynamics of only a limited number of cell types, we have yet to grasp the full complexity of global *in vivo* cell dynamics in normal developmental processes and disease. Here, we devise a computational method, digital cell quantification (DCQ), which combines genome-wide gene expression data with an immune cell compendium to infer *in vivo* changes in the quantities of 213 immune cell subpopulations. DCQ was applied to study global immune cell dynamics in mice lungs at ten time points during 7 days of flu infection. We find dramatic changes in quantities of 70 immune cell types, including various innate, adaptive, and progenitor immune cells. We focus on the previously unreported dynamics of four immune dendritic cell subtypes and suggest a specific role for CD103$^+$ CD11b$^-$ DCs in early stages of disease and CD8$^+$ pDC in late stages of flu infection.

**Keywords** cell quantification; deconvolution approach; dendritic cells; immune cell dynamics; influenza infection
**Subject Categories** Computational Biology; Immunology
**Mol Syst Biol. (2014) 10: 720**

## Introduction

An effective immune response requires the coordination and balance of hundreds of specialized immune cell subsets in the milieu of specific tissue content (Damjanovic *et al*, 2012). The holistic immune response is primarily determined by the dynamic physiological changes in each of the cell subsets, including proliferation, migration, differentiation, and transitions in cell activity. While it is clear that such physiological changes are essential for establishing appropriate immunological outcome, how these dynamic processes are orchestrated *in vivo* and what are the dynamics of each cell type during infection are still not fully understood.

Multiple studies have demonstrated the power of monitoring changes in the quantities of various immune cells to reveal their physiological changes and distinct functionality in health and disease (Newell *et al*, 2012; Brandes *et al*, 2013). A number of methods, such as fluorescence-activated cell sorting (FACS; Ibrahim & van den Engh, 2007), cytometry by time-of-flight (CyTOF; e.g. Bendall *et al*, 2011), and confocal/two-photon imaging (e.g., Stoll *et al*, 2002) have been developed and perfected along the years to address this important challenge. These are potent tools for immunology research and for monitoring changes in immune cell quantities, but are limited to a small number of cell subsets (Ho *et al*, 2011; Moltedo *et al*, 2011; Newell *et al*, 2012; Tate *et al*, 2012; Brandes *et al*, 2013) and require tissue destruction, thereby affecting cellular integrity and accuracy. Recent computational algorithms offer a parallel and powerful approach to infer the changes in cell quantities from gene expression data of a complex tissue. Most methods model each of the cell types independently (Nakaya *et al*, 2011; Josset *et al*, 2012; Parnell *et al*, 2012), but they fail when cell type cannot be easily distinguished—such as in the case of many related immune cell types. Complementary deconvolution approaches overcome this problem using a detailed model to resolve all cell types simultaneously (Lu *et al*, 2003; Wang *et al*, 2006; Abbas *et al*, 2009). However, such models contain many parameters (one for each cell type) and are therefore not scalable to a large number of cell types. Hence, current technologies are unsuitable for a holistic view of the dynamical

1 Department of Immunology, Weizmann Institute, Rehovot, Israel
2 Cell Research and Immunology Department, Tel Aviv University, Tel Aviv, Israel
3 Central Virology Laboratory, Ministry of Health, Public Health Services, Sheba Medical Center, Tel Hashomer, Ramat Gan, Israel
4 Faculty of Life Sciences, Bar-Ilan University, Ramat Gan, Israel
5 Department of Epidemiology and Preventive Medicine, School of Public Health, Sackler Faculty of Medicine, Tel-Aviv University, Tel-Aviv, Israel
*Corresponding author. Tel: 972 3 5302455; Fax: 972 3 530 24 57; E-mail: Michal.Mandelboim@sheba.health.gov.il
**Corresponding author. Tel: 972 3 6406945; Fax: 972 3 6422046; E-mail: iritgv@post.tau.ac.il
***Corresponding author. Tel: 972 8 9346974/5; Fax: 972 8 9345176; E-mail: ido.amit@weizmann.ac.il
†These authors contributed equally to this work.
§These authors contributed equally to this work.

changes occurring in hundreds of immune cells during normal physiology and disease.

Here, we present a novel algorithm called digital cell quantifier (DCQ), to infer global dynamic changes in immune cell quantities within a complex tissue (Fig 1). DCQ takes as input genome-wide transcription profiles of organs that were measured in two or more conditions (e.g., time points, perturbations). DCQ infers changes in cell quantities between the two conditions based on a cell surface markers motivated model. The outputs are interpretable hypotheses about changes in quantities of specific immune cell types between the two conditions. We employ three novel strategies to enable an accurate analysis for a large number of cell types. First, we modified the deconvolution approach into a regularized regression model to reduce the number of model parameters. Second, we based our analysis on a gold standard set of cell surface marker genes, used for over a decade in separating specific immune cells using FACS analysis. Finally, we learn an ensemble of models and use them to build a unified, robust solution.

We applied DCQ to follow the *in vivo* dynamics of 213 candidate immune cell types upon flu infection. Given detailed time series of RNA-Seq profiles from the lung tissue of influenza-infected mice, our analysis reveals significant changes in 70 immune cells, from progenitors (e.g., GMP, CMP, MEP) to various effector cells of both the innate and adaptive immune system. DCQ predicts known changes in cell type quantities with high accuracy, outperforming extant methods. Importantly, DCQ discerns closely related immune subtypes that have distinct changes in cell quantities, such as the differential dynamics of NKTs from different origins in the body. We validate our predictions of previously unreported changes in the quantities of four dendritic cell (DC) subtypes during influenza infection. We show that $CD8^+$ plasmacytoid DCs (pDCs) are recruited during the later phases of infection compared to $CD103^+$ $CD11b^-$ classical DCs (cDCs), suggesting a function for pDC as a cavalry to maintain long-lasting defense against influenza infection.

Our method opens the way to routine mapping of high-resolution temporal changes in each of hundreds of immune cell types within a tissue. We provide DCQ as a web-based software tool (http://www.DCQ.tau.ac.il), offering testable hypotheses about the dynamics and function of specific immune cells in normal physiological responses and disease.

## Results

### DCQ: an algorithm to infer global dynamics of immune cells from a complex tissue

To systematically decipher the *in vivo* cellular dynamics of the entire immune system during influenza infection, we devised a general and holistic computational approach to study the changes in quantities of immune cell subpopulations during the course of physiological response or disease (Fig 1). First, we extract the RNA from a complex tissue during the course of disease or physiological response (here, lung tissue during influenza infection) to "freeze" the tissue state and measure genome-wide gene expression profiles from each time point. We then load the genome-wide gene expression profiles into a novel algorithm we developed, called digital cell quantifier (DCQ), to computationally infer the global dynamics of

immune cell subsets during the course of disease (Methods; Fig 1). Finally, with a holistic view of immune cells dynamics, we use DCQ predictions to study critical immune cell subtypes that change in quantity during the course of the disease and dissect their activity during disease pathogenesis.

Since current deconvolution algorithms are not optimized to follow accurately the dynamics of dozens of immune cell types (Lu *et al*, 2003; Wang *et al*, 2006; Abbas *et al*, 2009), we developed a global digital cell quantifier (DCQ). DCQ takes as input (1) *differential expression data*—the observed change in expression of each gene *i* among two samples of a whole tissue (denoted $m_i$); and (2) an *immune cell compendium*—a collection of the prior knowledge about the mRNA concentration of each gene *i* in a cell type *j* (denoted $b_{ij}$), where many of the expressed genes are shared among different cell types. As an output, DCQ provides (*predicted*) *relative cell quantities* of cell types, namely the change in the amount of each cell type *j* before and after infection (denoted $c_j$).

Current deconvolution approaches (Lu *et al*, 2003; Wang *et al*, 2006; Abbas *et al*, 2009) model the change in the amount of a gene $m_i$ as the sum of relative quantities of many different cell types, each of which contributes a corresponding change in the total expression of the gene: $m_i = \sum_{j=1...p} c_j \cdot b_{ij}$, where all genes are affected by the same changes in cell quantities. The inferred parameters are the predicted relative cell quantities for each of the cell types (Supplementary Fig S1). A major weakness in such detailed deconvolution approaches is that they are limited to analysis of a small number of cell types, mainly because the number of cell types scales with the amount of parameters, which commonly leads to overfitting and strong biases due to the set of genes on which the analysis is applied.

To tackle these problems, DCQ combines three novel strategies that allow scaling up to hundreds of cell types that are needed for global view of immune dynamics. First, we modified the above deconvolution equation into a regularized regression called "*elastic net*" (Zou & Hastie, 2005), which combines both *l1* and *l2* penalties to penalize the model for a large number of parameters. This lowers the dimensionality of the search space, making DCQ more robust and scalable for a large number of cell types. Using simulated data demonstrates that elastic net regularization provides more robust results compared to alternative approaches (Supplementary Fig S2; Methods).

Second, we apply the approach to a pre-defined set of immune cell surface markers spanning all cell types under study. The gene set is comprised of the gold standard cell surface markers, used in immunology research to specifically separate (by FACS) all immune cells that are included in the DCQ immune cell compendium (Supplementary Tables S1 and S2; Supplementary Fig S1 and Materials and Methods). Our semi-supervised approach builds on extensive immunological knowledge and differs from previous approaches (Wang *et al*, 2006; Abbas *et al*, 2009), which selected genes that best discriminate between cell types regardless of their biological relevance. For each of these markers, we validated that its pattern of gene expression across cell types resembles its established intensity in FACS analysis on the cell surface of the various immune cell types (qualitative protein abundance is from Benoist *et al*, 2012; Kindt *et al*, 2007; Murphy, 2012; Supplementary Fig S3). Correspondence between gene expression and cell surface protein across cell types is calculated by a *t*-test score (Supplementary Table S2).

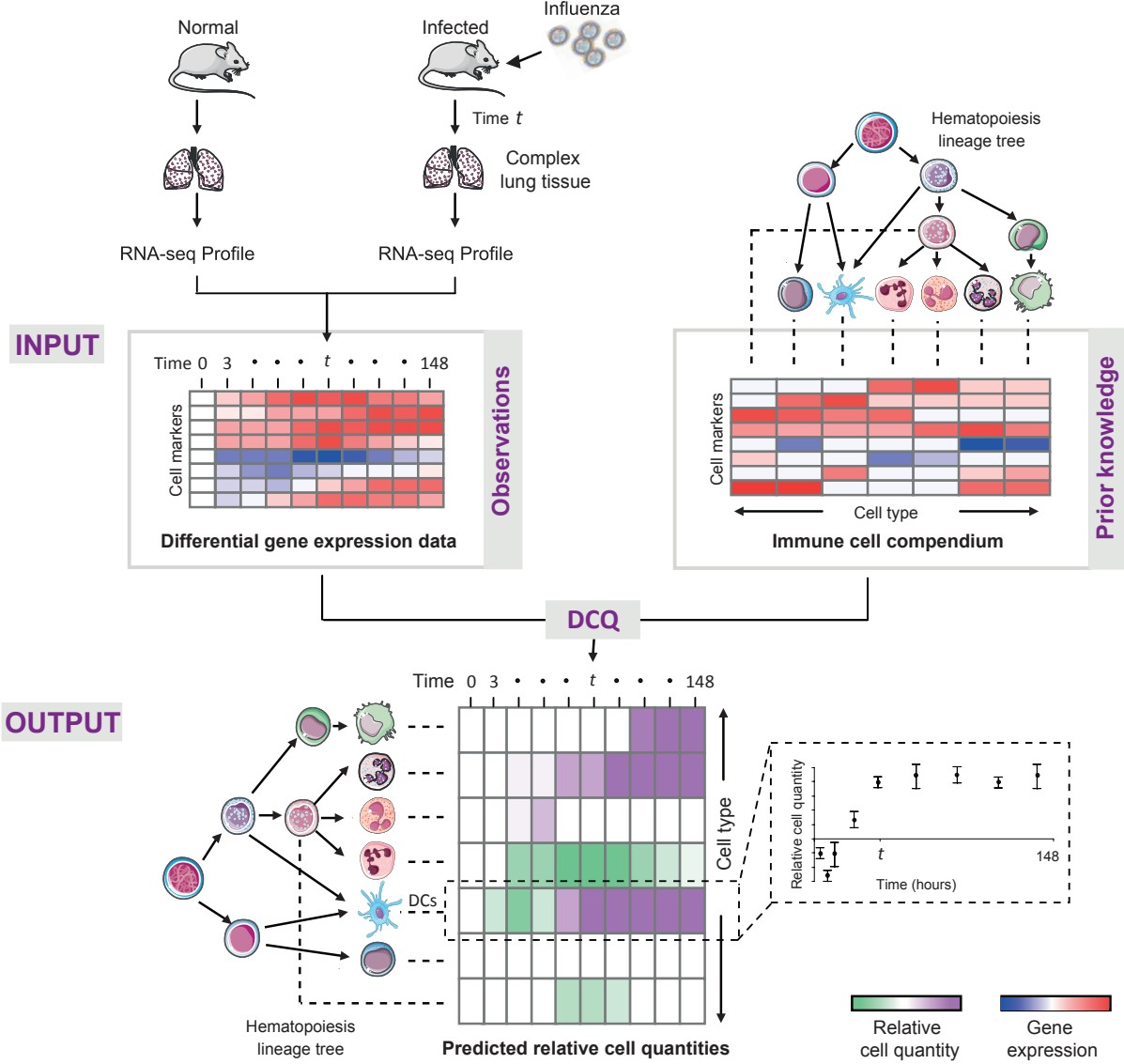

**Figure 1.    Overview of the digital cell quantifier (DCQ) algorithm.**
Our DCQ method takes two gene expression datasets as input: First (top left), differential genome-wide expression data from a complex tissue (here, lung), where rows are gene and columns are samples (here, time points during infection), high and low transcript level is color-coded in red and blue, respectively. Second (top right), a precompiled compendium of prior information about the abundance of each cell surface marker in each immune cell type (rows—markers; columns—cell types). Immune cell types are illustrated together with their hierarchical hematopoietic cell lineages. DCQ provides as output a matrix (bottom) of predicted relative cell quantities for each immune cell type (row) in each sample (column). Increase or decrease in cell quantity is color-coded in purple and green, respectively. Scatter plots (bottom right) exemplify the inferred amount of dendritic cells (y-axis) during the time course of infection (x-axis), where DC's quantity is reduced during the initial few time points and then elevated during latter time points. Standard deviations are calculated by DCQ based on an ensemble of alternative solutions (see Materials and Methods).

Finally, we devised an approach for evaluating the robustness of DCQ predictions. Rather than learning a single DCQ model, we infer an ensemble of models, each of which is based on a sample of 50% of the immune cell types. This allows us to calculate the significance (and standard deviation) of predicted relative cell quantities, referred to as a *robustness score* (Methods).

**DCQ performance in an *in vitro*-defined cell mixture**

Before applying DCQ in complex *in vivo* settings, we first confirmed its performance on a small number of cell types in a pre-defined controlled setting. To that end, we generated an *in vitro* complex cell mixture where the amount of each cell type is known. We isolated B cells, CD4[+] T cells, CD8[+] T cells, NK cells, and CD11c[+] DCs from mouse spleen. We mixed the isolated subsets of immune cells in various concentrations (from 1% to 10%, altogether ten different "tissue" samples) with a fixed high percentage of non-immune cells, generating pre-defined samples that closely resemble the immune dynamics in a complex tissue (Methods). In addition, we constructed an immune cell compendium for these samples by sequencing the RNA of each of the five cell subsets in isolation. As DCQ is designed for differential gene expression data, we further

created a reference sample of equal immune cell quantities and used it to calculate differential gene expression for these samples. Using these differential data, for each sample we compared (i) DCQ's predicted relative cell quantities and (ii) the input relative cell quantities, that is, the known input cell quantities versus the reference samples. We found high correlation between the known and predicted relative cell quantities (Pearson's $r = 0.90$, Fig 2A, B and Supplementary Table S3). In particular, high correlations are observed in each of the cell types separately ($r = 0.98, 0.66, 0.86, 0.92$, and $0.72$ for B cells, CD4$^+$ T cells, CD8$^+$ T cells, NK cells, and CD11c$^+$ DCs, respectively), corroborating DCQ ability to accurately predict dynamic changes in quantities of a small percentage of immune cell types within a complex cell mixture.

To further explore the optimal coverage of sequencing data on cell quantity output, we tested different depths of 3′-end RNA-Seq on our *in vitro* mix of several cell types (Methods). We observe that the accuracy increases with additional sequencing data, and saturated at a moderate depth of 2.5 million reads per sample (Fig 2C).

## Global immune cell dynamics during influenza infection

We next examined *in vivo* gene expression dynamics of influenza pathogenesis. To that end, C57BL/6 female mice were infected intranasally with $4 \times 10^3$ PFU of influenza PR8 virus. We measured, using RNA-Seq, the global gene expression dynamics in lung tissue at ten time points during a 7-day time course of infection, two infected individuals in each time point and four uninfected individuals as control (Supplementary Fig S4A; Methods). The gene expression is highly reproducible between two independent mice at the same time point (average Pearson's $r = 0.89$). Using quantitative PCR (qPCR), we confirmed the temporal profiles for representative genes (Supplementary Fig S4B; Methods). We also measured two indicators of disease progression for all animals, virus concentration in the lungs, and body weight loss (Supplementary Fig S5; Methods), which were reproducible across individuals and show only little variation between two independent mice at the same time points (Supplementary Fig S5).

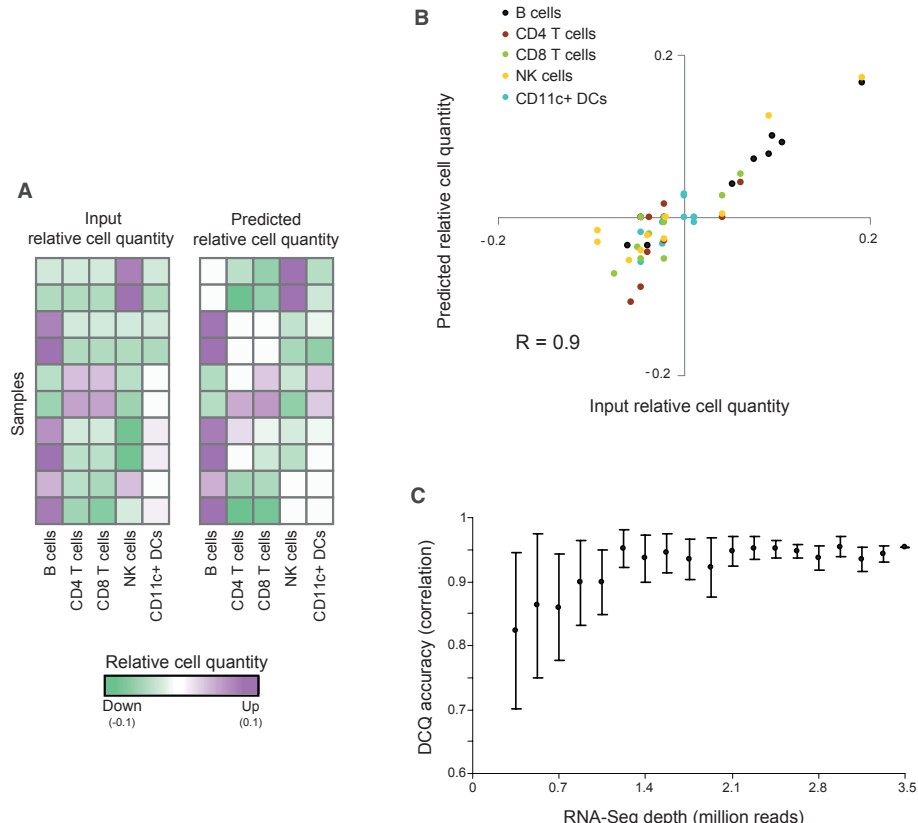

**Figure 2.  Digital cell quantification (DCQ) reconstruction of an *in vitro*-defined complex cell mixture.**

A   Performance analysis on ten samples generated using an *in vitro*-defined complex cell mixture (see Materials and Methods). The two matrices indicate the agreement among relative quantities that were inferred by DCQ (right) and the input relative cell quantities (left) for ten different experimental samples (rows), each of which involves five immune cell subsets (columns). Increase or decrease in cell quantity is color-coded in purple and green, respectively.

B   A summary of DCQ's predicted relative cell quantities (*y*-axis) and input relative cell quantities (*x*-axis) across all ten samples from a. The plot indicates the high correlation in each of the cell types (color-coded: black—B cells; brown—CD4$^+$ T cells; green—CD8$^+$ T cells; yellow—NK cells; cyan—CD11c$^+$ DCs).

C   The effect of RNA sequencing depth on DCQ performance, tested on the dataset from a. Accuracy of DCQ predictions (*y*-axis) are presented for various RNA sequencing depths (*x*-axis). Accuracy is evaluated as correlation between predicted and input ("true") relative cell quantities. Depicted are average of correlation and standard deviation over ten samples of each sequencing depth. The evident saturation with increasing depths implies that a sequencing depth of 2.5 million reads or higher is sufficient to provide high DCQ accuracy.

In agreement with previous reports (Rowe *et al*, 2010; Parnell *et al*, 2012; Pommerenke *et al*, 2012), we find that both our disease progression symptoms and gene expression data successfully capture the three main phases during influenza infection (Supplementary Figs S4–S6): First, the incubation phase (up to 26 h postinfection), characterized by an increase in virus particles in the lung, no sign of diseases in any of the animals, and a drastic repression of respiration and protein translation ($P < 10^{-8}$ and $10^{-9}$, respectively; Supplementary Fig S6A). Second, the disease progression phase (26–122 h), manifesting the onset of physiological phenotypes, high amount of virus particles, and a sustained increase in innate and adaptive immune responses, anti-viral and cell death genes ($P < 10^{-9}$, $10^{-8}$, $10^{-9}$ and $10^{-15}$, respectively). Third, the diseases outcome phase (122–148 h), characterized by either elimination of viral load and decrease in disease symptoms or continued reduction in body weight and death. The second and third phases show a significant over-representation of immune cell quantity and cell surface marker genes ($P < 10^{-15}$ and $10^{-9}$; 32 h postinfection, respectively; Supplementary Fig S6), suggesting dynamical changes in quantities of certain immune cells.

We sought to use DCQ to identify changes in immune cell types using the influenza-infected profiles. To that end, we compiled the prior compendium, consisting of 213 innate and adaptive immune cell types and their corresponding cell surface markers (Supplementary Tables S1 and S2; Materials and Methods). The compendium consists of both naïve and effector immune cell subsets that were isolated from 22 different tissues (e.g., spleen, liver, intestine) in both resting and activated immune states. Using this compendium and the influenza differential gene expression, we constructed a comprehensive map of the dynamic changes in quantities of 213 immune cell types during the course of influenza infection (Fig 3; Supplementary Table S4). Taking into account significant robustness scores in two consecutive time points, we identify a total of 69 *significantly changing cell types* (see Materials and Methods), 39 increasing and 31 decreasing, one cell type is both increasing and decreasing in different time points (Supplementary Table S5). In comparison, no significantly changing cell types were found in a permutation test (reshuffling the expression values of the cell surface markers among time points; Methods).

DCQ correctly predicts many known changes in immune cell quantities during infection. For example, DCQ infers a significant increase in the quantity of stimulated macrophages (MFs), but not resident MFs. Specifically, there is substantial increase of activated MFs: MHCII-F4/80$^{hi}$ CD115$^+$ and MHCII-F4/80$^{int}$CD115$^+$ MFs that were monitored at 5 days poststimulation with thioglycollate (denoted "MF.MHCII-F4/80$^{hi}$ Thio-5d" and "MF.MHCII-480$^{int}$ Thio-5d," respectively), and CD11c$^{-/lo}$Ser MFs at 3 days poststimulation with Salmonella ("MF.11cloSer, Salm-3d," carrying CD45$^+$ MHCII$^+$ CD11c$^{lo}$ CD11b$^+$ CD103$^-$ markers; Supplementary Fig S7A). Either no change or decrease is predicted for various subsets of steady-state MFs, including MF.CD103$^-$CD11b$^+$, MF.CD115$^{int}$ and MF.Siglec-5$^+$ (PI- CD45$^+$ MHCII$^+$ CD11c$^{hi}$ CD103$^-$ CD11b$^+$, B220$^-$ CD3$^-$ Ly$^-$6C$^-$ CD115$^{int}$ F4/80$^+$ and CD11c$^+$ MHCII$^-$ CD11b$^-$ CD103$^-$ SiglecF$^+$, respectively; Supplementary Fig S7A and Supplementary Table S1).

Monocytes, T cells, B cells, and NKTs provide additional examples for the ability of DCQ to identify previously known increase or decrease in quantities of immune cell types: The amount of various subsets of resident Ly$^-$6c$^-$ monocytes is decreasing during the incubation phase, but inflammatory Ly$^-$6c$^+$ monocytes show a marked increase during the progression and outcome phase, in agreement with previous literature (Gordon & Taylor, 2005) (Supplementary Fig S7B). As previously reported (Pommerenke *et al*, 2012), a drastic increase is inferred for the effectors and memory CD8$^+$ T cells but not for naïve subsets of CD8$^+$ T cells (Fig 3; Supplementary Fig S7C). Many pre-B, pro-B, germinal center, and naïve B cells remain stable or reduced during infection, whereas the amount of effector B cells, such as plasma and follicular-stimulated B cells, is increasing (denoted "B.Plasma_cells ST-7d" and "B.FO ST-6 h" in Fig 3, respectively). Notably, not only DCQ distinguishes cell types with distinct expression profiles, it also discerns between cell types with similar expression profiles (Supplementary Fig S8). This demonstrates that DCQ's predictions fit well with current knowledge of immune cell dynamics during infection.

Changes in cell quantity of NKT cells illustrate DCQ ability to relate cell types to their tissue of origin or migration from one tissue source to another. Overall, the immune cell compendium consists of 213 different immune cell types, 36 of them were isolated from more than one tissue. Using this compendium, DCQ can predict not only changes in quantities of a given cell type, but also specify whether the changing cell subset is typical to a particular body tissue. For example, the immune cell prior compendium consists of several different subpopulations of NKT cells that were isolated from spleen, liver, or thymus. DCQ inferred an increase in lung quantities of the spleen and liver NKT cells, but not the thymic NKT cells (Supplementary Fig S7D). This prediction holds for all subsets of NKTs under study, regardless of their levels of CD4, CD44, and NK1.1 markers, in agreement with the known proliferation and immune functionality of peripheral NKT cells (from liver and spleen), but not thymic NKTs (Godfrey *et al*, 2010). Notably, we cannot rule out the possibility that the distinct predicted NKT quantities in different tissues are due to different ways by which the cell types were isolated in our compendium. Yet, the apparent similarity of NKT expression profiles across spleen, liver, and thymus supports DCQ's ability to distinguish cell types from different origins (e.g., Supplementary Fig S8D).

As another example, resident Ly$^-$6c$^-$ monocytes show early transient reduction, followed by an increase in inflammatory Ly$^-$6c$^+$ monocytes (Supplementary Fig S7B). This prediction is supported by previous studies, indicating that resident monocytes migrate at early time points to periphery lymph nodes (hence the observed reduction in cell quantity), whereas the inflammatory monocytes are increasing due to differentiation and migration of monocytes from the blood compartment (Gordon & Taylor, 2005). The same results are predicted for monocytes that were derived in our compendium from blood, bone marrow, and lymph nodes, indicating that these subpopulations are closely related, and DCQ cannot split these cells apart with the current resolution of its input data and cell markers.

The effector CD8$^+$ T cells are a clear example of our approach to reveal the temporal process of immune cell activation in response to stimuli. Our prior compendium of immune cells consists of effector CD8$^+$ T-cell types following stimulations with Listeria at 12, 24, and 48 h, 6, 8, 10, and 15 days postinfection (Supplementary Fig S7C). As expected, DCQ infers increase in the quantity of 12-, 24-, and 48-h effector CD8$^+$ T cells and also an increase in 6 days postinfection, but no change is inferred for 8-, 10-, or 15-day effector CD8$^+$ T cells. This result exemplifies DCQ's capability to identify not only the immune cell type, but also its particular timing of activation state.

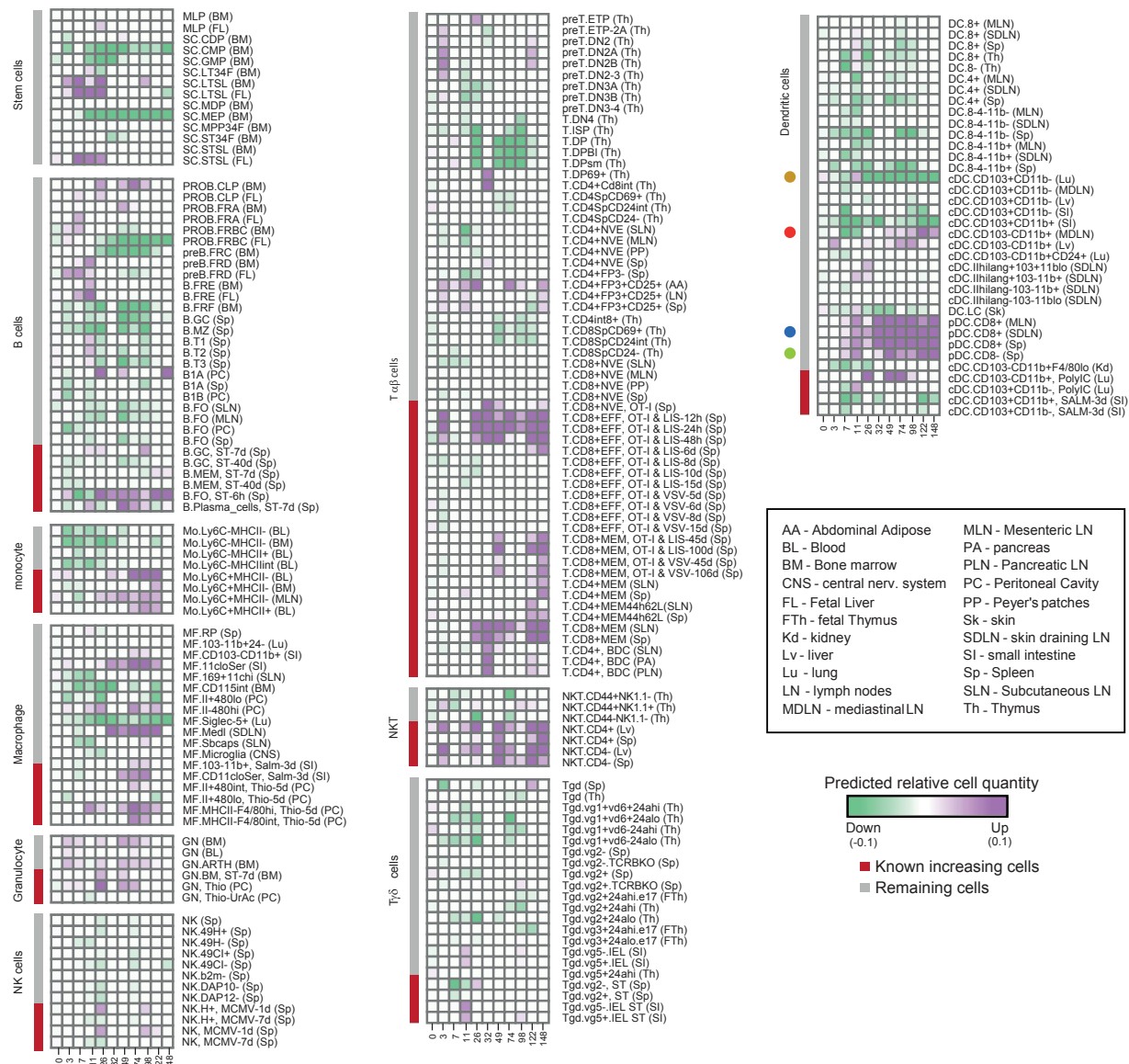

**Figure 3.  Digital cell quantification (DCQ) reconstruction of global immune cell dynamics during *in vivo* influenza infection.**
The immune dynamics map: Global dynamics in cell quantities (green/decrease, purple/increase in relative cell quantities) following influenza infection, predicted by DCQ at different time points (columns) for 213 different immune cell types (rows). Previously reported increase in cell types is marked in red (left, color bar). Each cell type heading is followed by the code of the tissue from which the cell type was isolated in the compendium. The box at the bottom right contains details for these abbreviations. Dendritic cells are shown at the top right panel and accompanied with four colored circles, indicating those subsets that were subject to FACS validations (see also Fig 5).

The reduction in the quantity of progenitor cells GMP, CMP, MEP and double positive (DP) CD4$^+$ CD8$^+$ T cells is also notable (Supplementary Fig S7E). Changes in the quantity of peripheral progenitors upon immune activation were previously reported (Massberg *et al*, 2007; Yeh *et al*, 2007; Johns *et al*, 2009), although not in the context of influenza infection. Although double positive (DP) undifferentiated T cells have been characterized in the peripheral blood and secondary lymphoid tissues (Zuckermann, 1999; Kenny *et al*, 2000; Nascimbeni *et al*, 2004; Bismarck *et al*, 2012), the decreased quantity of double positive (DP) undifferentiated T cells is previously unreported. In particular, DCQ predict the reduction in CD4$^+$ CD8$^+$ TCR$^{-/lo}$ CD69$^-$, CD4$^+$ CD8$^+$ TCR$^{-/lo}$ FSC$^{hi}$ and CD4$^+$ CD8$^+$ TCR$^{-/lo}$ FSC$^{lo}$ (referred to as T.DP, T.DPBl and T.DPsm in

Supplementary Fig S7E, respectively). There is no clear explanation for this observation, but these reduced quantities provide a potential mechanism by which peripheral progenitors collect information in the infected tissue before they differentiate or lead to the activation of other specific cells.

**Evaluation of DCQ accuracy using previously annotated cell types**

Next we focus on how well our learnt model of changes in cell quantities recapitulates established changes in cell quantities during infection. To that end, we constructed a literature-based list of 59 cell types, which are part of the prior compendium and also were previously reported as increasing in quantity during infection. This

list consists of various activated, effector and memory cell types and termed here "known increasing cell types during infection" or in short "known increasing" cell types (Fig 3; Supplementary Table S1). The remaining cell types are either progenitors and naïve cells, cell whose state is yet poorly characterized, and cells whose quantity likely decreases during infection. Of 39 significantly increasing cell types based on DCQ's prediction, 24 (61%) are known increasing cell types (hyper-geometric $P < 10^{-6}$). Principle component analysis of DCQ's predictions further illustrates the distinct distribution of known increasing cell types compared to the remaining cell types ($P < 10^{-9}$, Kolmogorov–Smirnov test; Fig 4A). To determine the performance of DCQ, we used this classification of known increasing versus remaining cell types as a gold standard, and evaluated the tradeoff between the true- and false-positive rate across various DCQ robustness cutoffs (Methods).

Using these criteria, we investigated the performance of five alternative marker selection methods. These are referred to as "All," "Max," "Max ratio," "Random," and our own "FACS-based" approach (Methods) and consist of 276, 696, 366, 61, and 61 marker genes, respectively. We find that using the elastic net algorithm, the FACS-based markers outperform alternative strategies (Fig 4C). For example, in a false-positive rate of 0.2, the FACS-based subset of markers attains a true-positive rate of 0.64, whereas the true-positive rate of alternative subsets ranges between 0.29 and 0.51.

We next aimed to test various regression methods, including a non-regularized linear regression and regularized regression with $l1$ penalty (lasso; Tibshirani, 1996) or a combination of $l1$ and $l2$ penalties (elastic net; Zou & Hastie, 2005; Supplementary Text 1). Each of the (regularized or non-regularized) regression methods is tested, if possible, across the five marker selection methods. We find that using any of the above five methods for marker selection, elastic net outperforms lasso and non-regularized regression (Supplementary Fig S9). For example, using FACS-based selection of markers and given a false-positive rate of 0.15, DCQ and lasso provide a true-positive rate of 0.57 and 0.3, respectively.

Finally, we compared DCQ to recent advanced cell quantification methods, which use different combinations of computational prediction and marker selection methods, including (i) a non-regularized linear model with the "Max" group of 696 markers [as in Lu et al (2003)], (ii) a non-regularized linear model with the "Max ratio" group of 366 markers [as in Abbas et al (2009)], and (iii) an hypergeometric enrichment tests, using a list of 20 genes that are characteristic of each cell type [as in Nakaya et al (2011)]. Compared to these robust methods, the DCQ algorithm shows a better true- to false-positive rate tradeoff (Fig 4B). For example, for a false-positive rate of 0.3, DCQ attained a true-positive rate of 0.75, whereas true-positive rate of alternative methods ranges between 0.33 and 0.42 (Fig 4B). Similar results were obtained when only a fraction of the selected markers were used (Supplementary Fig S10).

Taken together, these results provide strong support for the accuracy of DCQ in predicting a global map of changes in immune cell types during the course of infection.

### Heterogeneity in DC dynamics highlights the role of DC subtypes in influenza infection

One particular interesting prediction of DCQ is heterogeneity in quantity changes in DCs [Fig 3, bottom right panel; Supplementary Table S5; (Geissmann et al, 2010)]: Two subtypes of plasmacytoid dendritic cells (pDCs) undergo robust early increase during infection (CD8$^+$ and CD8$^-$ pDCs); the quantity of one conventional/classical DC (cDC) is slightly increased and then reduced (CD103$^+$ CD11b$^-$); and one cDC subset (CD103$^-$ CD11b$^+$) manifests a late increase in cell quantity. To validate these predictions, mice were infected for 0, 24, 72, and 120 h (two mice per time point), and individual cells from all four DCs populations (CD8α$^+$ or CD8α− pDC and CD103$^-$ CD11b$^+$ or CD103$^+$ CD11b$^-$ cDC) were FACS-sorted (Methods) to measure their dynamics during the infection process. Consistently with DCQ predictions, both pDC subtypes show a substantial increase in cell quantities. For example, only 62 ± 10 CD8α$^-$ pDCs reside in lung prior to infection. This number increases to 348 ± 8 cells at 24 h postinfection and further increases to 9513 ± 5002 at 72 h postinfection and to 14,303 ± 1251 at 120 h postinfection (Materials and Methods, Fig 5A and Supplementary Table S6). Furthermore, our analysis correctly quantified the complex changes in CD103$^-$ CD11b$^+$ whose amount also increases, but at a later time points (Fig 5A). Unlike the pDCs, the quantity of CD103$^-$ CD11b$^+$ at 24 h is still similar to the quantity prior to infection (448 ± 178 cells and 338 ± 70 cells before and at 24 h postinfection, respectively). At 72 and 120 h, we observe a substantial increase in cell quantity (7563 ± 2329 and 17,907 ± 512 cells), in agreement with our predictions.

To better understand the physiological role of these differential dynamic changes in the DCs, we measured the genome-wide RNA expression of all four DC subpopulations from the lung of influenza-infected mice at four time points following infections (two mice per time point). As expected, we observe a marked difference in the genes that are expressed in each DC subtype compared to other cell types and compared to the entire lung tissue (Fig 5B). Analyzing the data closely, we observe a large difference in specific chemokines expressed by the DC subtypes. For example, the two cDC subtypes express high levels of CCL22 and CCL17 (Fig 5C, Supplementary Fig S11), ligands of the chemokine receptor CCR4 and potent chemoattractants for B and T cells (Luster et al, 2005), suggesting that each DC subtype interacts with a different immune niche and has a different role in the response.

In agreement, there is a substantial difference in the anti-viral response of the various DC subtypes (Fig 5D). CD103$^+$ CD11b$^-$ cDCs respond early (within the first 24 h of infection), but then enter an exhaustion-like state, characterized by the reduction in the intracellular anti-viral responses and cytokine production at 3 days postinfection. CD8α$^+$ pDCs, on the other hand, respond slower but maintain a long-lasting anti-viral response, even 5 days after infection, possibly because they do not enter an exhaustion-like state or constantly are replaced by new incoming pDC from other tissues. Taken together, the combined action of pDC and cDC subsets maintain together a rapid, prolonged anti-viral response of DCs in lung.

## Discussion

The main characteristics of the immune response and various immune-related disease are increased vascular permeability, cellular infiltration by chemotaxis, and other form of cell homing. The mobilization of immune cells from one organ to another, in addition to their differentiation and proliferation under specific cues, lies at the

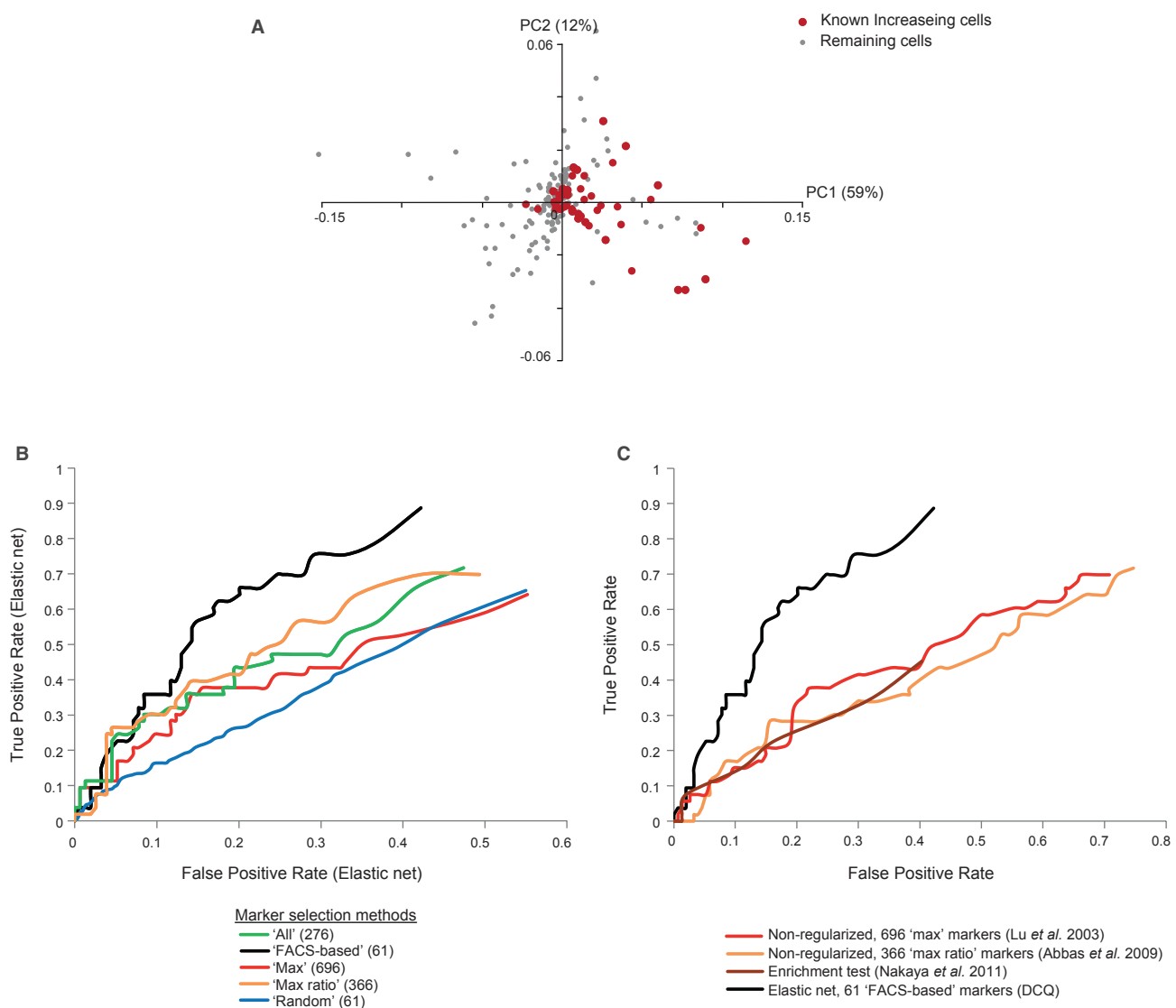

**Figure 4. Digital cell quantification (DCQ) correctly predicts changes in most known increasing cell types during influenza infection.**

A      Principle component analysis (PCA) of DCQ's predicted relative cell quantities. The PCA was applied on the profiles of predicted relative cell quantities for each cell type, at ten time points during influenza infection. Shown is a scatter plot of each cell type for the first two principle components PC1 and PC2. Red, cell types that were previously reported as increasing in quantity during infection; gray, the remaining cell types.

B, C  Comparison of performance. False- (*x*-axis) and true-positive (*y*-axis) rates of DCQ predictions. Rates are calculated for comparing predicted increase in cell quantities versus the known increasing cell types. In (B), we compare five alternative methods for selecting gene markers. In (C), we compare DCQ to several alternative computational cell quantification approaches, each consists of a different mathematical formulation and a different set of markers. The plots suggest the superiority of DCQ, and in particular its FACS-based selection of markers, over extant methods.

heart of the ability of the immune system to detect and mount a precise counter response to different pathological conditions (Murphy, 2012). Discovering the dynamics of the immune system is a difficult problem, requiring reliable resolution and simultaneous quantification of hundreds of immune cell types. Current studies, however, are typically limited to the investigation of only a few cell types simultaneously with limited characterization of the synchronization among cell types (Brandes *et al*, 2013; Pang *et al*, 2013). Here, we designed a digital cell quantifier that offers unique systematic quantification of immune dynamics in a global manner. Notably, DCQ allows simultaneous prediction of over 200 immune cell types and

can discriminate between closely related immune subtypes (e.g. NKTs from different origins in the body) and different levels of activity.

A key advantage of our method is its ability to generate detailed testable hypotheses concerning the role of specific immune cells under particular conditions. We offer experimental results supporting four of our computationally generated hypotheses, including two cDCs and two pDCs subpopulations. Transcriptome analysis suggests that different DC subsets acquire different immunological roles during the influenza infection, such as (i) different immunological niches, via expression of specific chemokines by the cDC

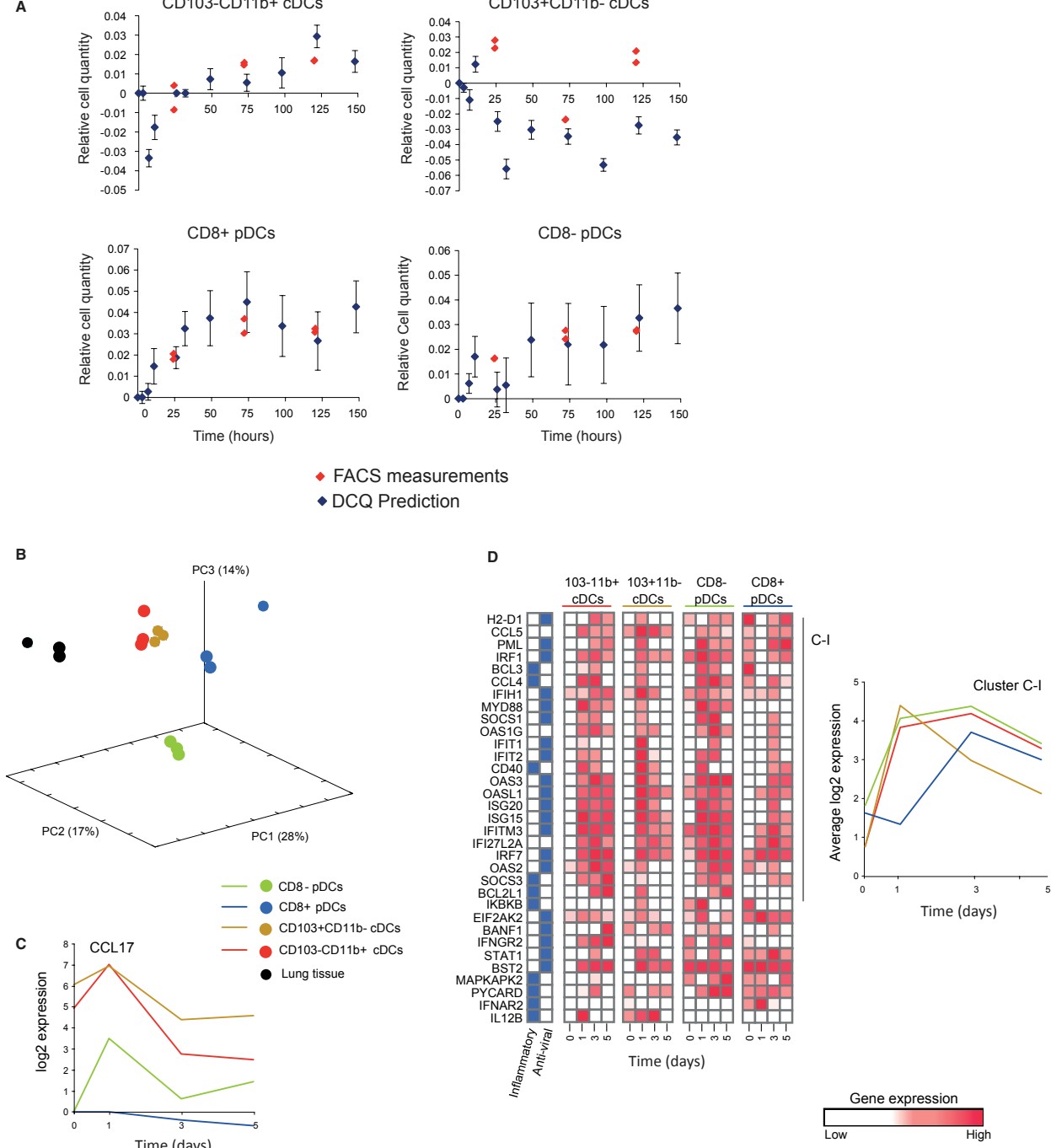

**Figure 5. Heterogeneity in DC dynamics highlights the distinct role of DC subtypes in influenza infection.**

A   Validation of cell quantity dynamics of four dendritic cell subtypes. Dendritic cell quantities (*y*-axis) as predicted by digital cell quantification (DCQ) (blue diamonds) and measured by FACS (red diamonds) for two mice at each time point (*x*-axis). The plot indicates the high correlation between DCQ predictions and FACS validations.

B   PCA analysis of the genome-wide transcriptional responses to influenza infection. The PCA was applied on RNA-Seq expression values that were profiled on isolated DCs (normalized by the respective values before infection). Shown are three time points (1, 3, and 5 days) for each of the DC subsets (blue, CD8[+] pDC; green, CD8[−] pDC; red, CD103[−] CD11b[+] cDCs; brown, CD103[+] CD11b[−] cDCs; black, lung tissue).

C   Expression of CCL17 (*y*-axis) at four time points during influenza infection (*x*-axis) for four DC subsets (color-coded as in B).

D   Gene expression profiles of four DCs populations during influenza infection in lung. Left: Shown are the (log2 ratio) expression levels of selected genes (rows) at three time points (columns) for four isolated subsets of DCs relative to control subsets before infection; Z-normalized per row. Previously reported inflammatory and anti-viral genes in dendritic cells are marked in left (blue, Amit *et al*, 2009; Gat-Viks *et al*, 2013). Cluster C–I is highlighted. Right: average expression (*y*-axis) at each time point (*x*-axis) of genes in clusters C-I, for the four isolated DC subsets (color-coded as in B).

versus pDC (e.g., CCL17, CCL22); (ii) different temporal timing of activation allowing to mount rapid yet long-lasting anti-viral response rather than an untimely exestuation; and (3) differential expression of cell subset-specific anti-viral proteins, via the activation of different pathways. Our other hypotheses of immune cell dynamics during influenza infection have not yet been tested and merit further investigation.

Our method opens the way to many different directions of future research. One exciting possibility is that DCQ's prediction of cell dynamics, based on early gene expression in the blood or tissue of patients, can be used for disease prognosis and individual classification. For this to happen, it will be important to adjust the method for human samples and construct a prior compendium of immune cell types and surface markers in humans. Second, our method, with some modifications, will be applicable for other cell types and niches, such as different cancers and brain regions. Third, characterizing coordinated changes in cell quantities of several immune cell types might reflect a common underlying mechanism, thus providing important information of clinical relevance. For example, a few such "co-increasing" cell types might migrate together due to the same chemotaxis agent or proliferate in an orchestrated way in response to a similar combination of cytokines. While our method substantially outperforms extant methods for the case of many cell types, it can potentially be even further improved with alternative sets of markers that are specifically tailored for a given sample (rather than using a common pre-defined set of markers for all samples). Notably, DCQ cannot discern different immunological mechanisms (e.g., differentiation, migration); instead, its predicted relative quantities provide promising hypotheses for further mechanistic investigations.

Overall, our method provides a clear global view of the dynamics of immune system in a complex tissue and suggests hypotheses concerning the roles of these cells within organs in the body. It is our belief that applying DCQ to dynamics of both normal and pathological conditions may lead to important new insights toward understanding the complex roles of various immune cell types and suggest novel targets for clinical intervention.

# Materials and Methods

### Ethics statement

All animal work has been conducted according to relevant national and international guidelines.

### Virus

Mouse-adapted PR8 virus, influenza A/Puerto Rico/8/34 (A/PR/8/34, H1N1), was persistently grown in hen egg amnion, and influenza effective titer was quantified.

### Viral infection of mice

Female mice C57BL/6J (5 weeks of age) were anesthetized with isoflurane and were inoculated intranasally with 50 μl of diluted virus. The infected animals were observed for reduction in weight and were sacrificed on 3, 7, 11, 26, 32, 49, 74, 98, 122, 148 h

postinfection, two animals at each time point, together with a group of four uninfected mice. The lung organ was removed and transferred immediately into RNAlater solution (Invitrogen).

### RNA isolation

For RNA isolation, each organ was cut into small pieces in the presence of QIAzol and homogenized using SPEX CertiPrep homogenizer, and total RNA was extracted using miRNeasy Mini Kit (Qiagen). The RNA integrity number (RIN) was determined using the TapeStation System (Agilent Technologies). Quantity was determined by Qubit Fluorometric Quantitation (Life Technologies).

### Quantitative PCR

Total RNA was reverse-transcribed to cDNA using high capacity cDNA reverse transcription kit (Applied Biosystems). qPCR was performed with LightCycler480 SYBR Green I Master Mix (Roche) in triplicate using either GAPDH or Actb genes for normalization. Primers used for DNA amplification in the PCR assays are presented in Supplementary Table S7.

### Preparation of RNA sequencing libraries

For the preparation of RNA-Seq libraries, total RNA was fragmented into average size of 300 nucleotides by chemical heat (95°C) treatment for 4:30 min (NEBNext Magnesium RNA Fragmentation Module). The 3′ polyadenylated fragments were enriched by selection on poly dT beads (Dynabeads Invitrogen). Strand-specific cDNA was synthesized using a poly T-VN oligo (18 T) and Affinity Script RT enzyme (Agilent). Double-strand DNA was obtained using Second strand synthesis kit (NEB). DNA ends were repaired using T4 polynucleotide kinase and T4 polymerase (NEB-Next). After the addition of an adenine base residue to the 5′ end using Klenow enzyme (NEB-Next), a barcode Illumina compatible adaptor (IDT) was ligated to each fragment. The washed DNA fragment was amplified by PCR (12 cycles) using specific primers (IDT) to the ligated adaptors. The quality of each library was analyzed by TapeStation (Agilent).

### Pre-processing of RNA-Seq data

All reads were aligned to the mouse reference genome (NCBI 37, MM9) using the TopHat aligner (Trapnell et al, 2009). The raw expression levels of the genes were calculated using Scripture (Guttman et al, 2010), an ab-initio software which reconstruct transcriptomes. Normalization was done using DESeq (Anders & Huber, 2010) based on the negative binomial distribution and a local regression model. For the complex lung tissue data, we next applied a log2 transformation and normalized each entry by the gene's average (across the four steady-state values) and standard deviation during infection. No additional data transformation was applied to integrate this data with the compendium of cell types. Various sequencing depths were generated by sampling a fraction $f$ of reads at random ($f = 0.01$–$0.95$, Fig 2C) and then re-applying the above pre-processing pipeline on this sample of reads.

## Enrichment analysis of biological functions and pathways

For pathways and functional analysis (Supplementary Fig S6A), we collected cellular pathways from REACTOME (Croft *et al*, 2011) and augmented it with immune-related categories from the IPA (Systems, Mountain View, CA, USA) database. Function and pathway enrichments in a profile of a certain time point were calculated using a Wilcoxon test *P*-value. To map the regulatory entities acting during influenza infection (Supplementary Fig S6B-D), we used a previously assembled network of 1550 regulators and 409,000 regulator-target gene connections (Yosef *et al*, 2013). The control of a regulator at a given time point is determined by the enrichment of its set of putative targets in the up- or down-regulated genes at the relevant time point (a Wilcoxon test). All Wilcoxon *P*-values were adjusted for multiple testing with an FDR correction ($P < 0.001$).

## The DCQ algorithm

The DCQ method takes as input (i) an immune cell compendium, consisting of transcriptional profiles of isolated immune cell subsets, taken from various tissues, stimulations, and time points; (ii) a collection of immune cell surface markers designed to discriminate between the immune cell types included in the compendium; and (iii) differential gene expression profiles (e.g., from RNA-Seq data) of a test versus reference samples. DCQ predicts, for each cell type and time point, a "relative cell quantity" measure, namely the change in cell quantity between a sample at the relevant time point and a sample at steady state. DCQ applies elastic net regularization (Zou & Hastie, 2005), which relies on two model parameters (lambda.min.ratio and $\alpha$; Supplementary Text 1). In this work, we used the glmnet R package (Friedman *et al*, 2010) with the parameters $\alpha = 0.05$, lambda.min.ratio = 0.2. The source code is provided in Supplementary Text 2 and in http://www.DCQ.tau.ac.il.

To evaluate the robustness of the *predicted relative cell quantities*, we generated 100 modified versions of the prior compendium, each of which includes a random collection of only 50% of the cell types, and apply DCQ on them. This way, we collected an ensemble of 100 relative quantity solutions, indicating how modification of the prior compendium can affect the model. Standard deviations were calculated across this ensemble of relative quantities. The significance of a predicted change in quantity, called *robustness score*, is assessed by evaluating whether the sample of relative quantities is significantly different from zero (a *P*-value score). Here, *significantly changing cell types* are those whose $-\log_{10}$ *t*-test *P*-value score is higher than a certain robustness cutoff during at least two consecutive time points (Supplementary Table S5). The application of *t*-test rather than an a-parametric test such as Wilcoxon is not theoretically justified, but provides a better performance in practice.

## The immune cell compendium

The immune cell compendium is collected from two complementary datasets (Benoist *et al*, 2012; Kaji *et al*, 2012) and includes 14 stem cells, 30 B cells, 89 T cells (51 $\alpha\beta$ T cells, 22 $\gamma\delta$ T cells, and 16 activated (effector and memory) T cells), 12 NK cells, 8 monocytes, 18 macrophages, 36 DCs, 6 granulocytes, and additional 10 stromal cells, resulting in a total of 223 cells, 213 of them are immune cell types (Supplementary Table S1). This study is focused on these 213

immune cell types, 45 of them were profiled under pathogen-like stimulation (22 T cells, 4 NK cells, 6 macrophages, 4 DCs, 3 granulocytes, and 6 B cells; Supplementary Table S1). The transcriptional profiles were processed as follows: First, we applied RMA normalization (Irizarry *et al*, 2003). Next, $\log_2$-transformed profiles of the same cell type were averaged. Finally, for each given gene, all its values were normalized by its median and standard deviation.

## *In silico* simulation

To investigate DCQ on simulated data, we constructed pairs of reference and test samples, generated at random as follows: The cell types were a-priori divided into 58 different groups according to the known structure of the hematopoietic lineage tree (Jojic *et al*, 2013). Each reference sample consists of twenty groups of cell types that were randomly selected out of the 58 groups of cell types. As a first step, each group was assigned a starting fraction (0.05). The test sample is then generated by modifying its respective reference sample: increasing the fractions of ten groups of cell types while decreasing fractions of ten other cell types. To simulate noise effects, we added an error component that is sampled from a gaussian distribution whose standard deviation is the average of the standard deviation among the replicates of cell types in the compendium. The gene expression of a simulated sample is calculated by averaging across the profiles of the selected cell types (using the prior compendium with noise effect), weighting profiles by their simulated fractions. In order to compare DCQ and lasso performance with the non-regularized regression, we used a collection of 276 known cell surface markers that are commonly used to characterize the specific cell types in our analysis, and not the regular FACS-based markers. This list was obtained using the FACS-based 61 genes and additional genes that characterize these immune cell types (Kindt *et al*, 2007; Murphy, 2012).

## An *in vitro*-defined complex cell mixture

To investigate the performance of DCQ in a controlled setting, we mimicked the tissue complexity by generating an *in vitro* complex cell mixture, where the amount of each cell type is known. Spleens from C57BL/6J female mice were placed inside 70-μm cell strainers (BD Falcon) on petri dishes. Ice-cold RPMI with 10% FBS was added and the spleens were ground with the cap of a 3-ml syringe. The isolated cells were transferred to a 15-ml tube and centrifuged, and their red blood cells were lysed using ammonium chloride solution. Cells were filtered through 70-μm strainers and resuspended in FACS buffer. FC receptors were blocked with anti-mouse CD16/CD32, washed with FACS buffer, and stained for sorting. Before sorting, cells were filtered again through 70-μm cell strainers. Cells were stained with a subset of the following antibodies depending on sorting needs: PE/Cy7-conjugated anti-CD19 (clone 1D3), APC/Cy7-conjugated anti-CD45R (clone RA3-6B2), APC-conjugated anti-CD11c (clone N418), FITC-conjugated anti-TCRβ (clone H57-597), efluor 405-conjugated anti-NK1.1 (clone PK136), APC-conjugated anti-CD8α (clone 53-6.7), PE-conjugated anti-CD4 (clone RM4-5). Cells were gated for size, singlets and then by positive and negative markers: B cells were CD45R[+], CD19[+], TCRβ[−], CD11c[−]; CD4 T cells were TCRβ[+], CD4[+], CD45R[−], CD8[−]; CD8 T cells were TCRβ[+], CD8[+], CD45R[−], CD4[−]; NK cells were NK1.1[+],

TCRβ⁻, CD19⁻, CD45R⁻, CD11c⁻; DCs were CD11C⁺, CD45R⁻, CD19⁻, TCRβ⁻. All antibodies were purchased from Biolegend and eBioscience. Isolated B cells, CD4⁺ T cells, CD8⁺ T cells, NK cells, and CD11c⁺ DCs were mixed in various concentrations (from 1% to 10%, altogether 10 different "tissue" samples) or in isolation with a fixed 50% concentration of mouse fibroblast cells (representing tissue mass). RNA was extracted from these mixed tissues and RNA-Seq libraries were generated, sequenced, and quantified (Methods).

### Influenza infection analysis

DCQ is designed to imitate the standard experimental procedure for quantifying immune cell subsets. It therefore exploits the set of 59 cell surface markers that were used to isolate each of the cell types included in the compendium. Twelve of these markers do not appear either in the compendium or in our influenza lung RNA-Seq data and are therefore excluded from the analysis and substituted with alternative thirteen markers. In total, 61 different cell surface markers were utilized to isolate at least one of the cell types. These are called the "FACS-based markers" (Supplementary Table S2).

We compared the FACS-based markers to five alternative marker selection methods: (i) All cell surface markers—a collection of all 276 well-established cell surface markers that are commonly used to characterize the cell types in this study. The list is constructed based on two general literature references (Kindt *et al*, 2007; Murphy, 2012) and includes the 61 FACS-based markers. (ii) Following Lu *et al* (2003), choosing a set of 696 genes with the highest expression variability among cell types. (iii) Following Abbas *et al* (2009), choosing the set of genes that best distinguish between cell types. Selected genes are those with highest expression ratio. Ratios are calculated between each two neighboring cell types that are ranked based on the expression of a candidate gene (366 markers). Finally, (iv) random markers—a random set of 61 genes, selected among the collection of different genes that were profiled in the immune cell compendium dataset. We called these groups "All," "Max," "Max ratio," and "Random," respectively.

True- and false-positive rates are evaluated by comparing the group of significantly increasing cell types to a gold standard set of 59 cell types with a documented increase in quantity during infection (Supplementary Table S1). The gold standard set consists of (i) all cell types that were stimulated before profiling (45 cell types), (ii) cell types that are referred to as "inflammatory" or "effectors" by either the Immgen consortium (Benoist *et al*, 2012) or by a general immunological reviews (Godfrey *et al*, 2004; Gordon & Taylor, 2005; 20 cell types). To avoid biases, we do not include extant knowledge from specific (possibly contradicting) publications and therefore do not construct a similar gold standard set of decreasing cell types.

Curves of true- and false-positive rates were generated across varying robustness cutoffs. The area under this curve is an estimator of the *quality of a cell quantity prediction*. In contrast, the *quality of a marker prediction* is evaluated by a leave-one-out cross-validation while summing the mean squared prediction error over all the time points. Based on the clear tradeoff between the quality of marker prediction and quality of cell type prediction, we used the parameters lambda.min.ratio = 0.2 and $\alpha = 0.05$ that balance between those objective functions (Supplementary Fig S12A). Thus, our selected parameters do not provide the best TPR and FPR that can be attained. In principle, similar or even better results and

performance could be obtained for lambda.min.ratio = 0.08–0.5 and $\alpha = 0.01$–0.1 (Supplementary Fig S12B) and when applying an additive or multiplicative scaling of the differential RNA-Seq data (Supplementary Fig S13).

DCQ was applied on the lung data with robustness cutoff = 20. Permutation tests were performed by running DCQ on 10 permuted gene expression datasets and identifying significant cell types (with robustness cutoff = 20) in each of these permuted datasets. Permuted datasets were generated by reshuffling the expression values of the cell surface markers among time points. This way we maintained the markers and the correlations between the cell types in the compendium, while disrupting the correlation between markers and the order of time points.

### Fluorescence-activated cell sorting

For sorting dendritic cells from lungs, the lungs from infected and control uninfected C57BL/6J mice were immersed in cold PBS, cut into small pieces in 5 ml DMEM containing 10% bovine fetal serum, the cell suspensions were grinded using 1-ml syringe cup on a 70-µm cell strainers (BD Falcon). The cells were washed with ice-cold PBS. Remaining red blood cells were lysed using ammonium chloride solution (Sigma). Cells were harvested and immersed 1 ml FACS buffer [PBS+2% FBS, 1 mM EDTA]; Fc receptors were blocked with anti-mouse CD16/CD32, washed with FACS buffer, and divided into two tubes for sorting cDCs and pDCs.

For sorting cDC, the cells were stained with antibodies against multiple surface antigens: Percp cy5.5-conjugated anti-CD45 (clone –F11), APC-conjugated anti-CD11c (clone N418), PB-conjugated anti-I-A/IE (clone M5/114.15.2), PE-conjugated anti-CD103 (clone 2E7), and FITC-conjugated anti-CD11b (clone M1/70). The cDCs were identified as CD45⁺, CD11c^high and MHC-II⁺ and were gated as CD103⁻ CD11b⁺ and CD103⁺ CD11b⁻.

For pDC sorting, the cells were stained with the following antibodies: Percp cy5.5-conjugated anti-CD45 (clone –F11), APC-conjugated anti-CD11C (clone N418), APC CY7-conjugated anti-CD45R/B220 (clone RA3-6B2), PE-conjugated anti-PDCA-1 (clone 129c1), and PE CY7-conjugated anti-CD8α (clone 53-6.7). The pDCs were identified as CD45⁺, CD11C^intermediate, B220⁺, PDCA-1⁺ and gated as CD8α⁺ and CD8α⁻.

Flow cytometry was performed using SORP FACSAriaII Flow Cytometer (Becton Dickinson), and data were analyzed using WinMDI 2.8 software. Sorted dendritic cells were subject to RNA-Seq profiling.

### Accession codes

The complete RNA-Seq datasets are available from the Gene Expression Omnibus (GSE49934).

**Supplementary information** for this article is available online: http://msb.embopress

### Acknowledgements

Research in IA laboratory is supported by a grant from the European Research Council (309788), the Israeli Science Foundation (1782/11), the Human Frontiers Science Program, and a Career Development Award. Research in IGV laboratory is supported by the Israeli Centers of Research Excellence (I-CORE) Gene Regulation in Complex Human Disease, Center No 41/11, The Israeli

Science Foundation (1643/13) and the Edmond J. Safra Center for Bioinformatics at Tel-Aviv University (YS). IGV is a Faculty Fellow of the Edmond J. Safra Center for Bioinformatics at Tel-Aviv University and an Alon Fellow.

## Author contributions

ZA, YS, MM, IA, and IG-V conceived and designed the study. ZA, LV, HK-S, TM, and EM conducted the experiments. IG-V and YS conceived computational methods. YS developed, implemented, and analyzed the computational method. ZB-I and ED participated in the implementation of computational methods. ZA, YS, IG-V, and IA wrote the manuscript with input from all authors.

## Conflict of interest

The authors declare that they have no conflict of interest.

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
