## [Review Process File · Molecular Systems Biology]

Digital cell quantification identifies global immune cell dynamics in Influenza infection

Zeev Altboum, Yael Steurman, Eyal David, Zohar Barnett Itzhaki, Liran Valadarsky, Hadas Keren-Shaul, Tal Meninger, Ella Mendelson, Michal Mandelboim, Irit Gat-Viks, Ido Amit

Corresponding author: Ido Amit, Weizmann Institute of Science

Review timeline:

Submission date:	31 October 2013
Editorial Decision:	23 December 2013
Revision received:	22 January 2014
Editorial Decision:	27 January 2014
Revision received:	29 January 2014
Accepted:	30 January 2014

Editor: Thomas Lemberger

Transaction Report:

1st Editorial Decision

23 December 2013

Thank you again for submitting your work to Molecular Systems Biology. We have now heard back from two of the three referees who agreed to evaluate your manuscript. While we are still waiting for the report of the third reviewer, I prefer to make a preliminary decision now rather than delaying further the process. As you will see from the reports below, the present referees find the topic of your study of potential interest. They are however only cautiously supportive and they raise, however, substantial concerns on your work, which should be convincingly addressed in a major revision of the present study.

The points made by the reviewers are very clear and refer to the need of additional analysis and clarifications. The method should also be better described and a more detailed comparison to previous approaches should be performed.

On a more editorial level, the data generated and analyzed in this study --expression from immune cell compendium and expression during infection time course -- should be deposited in an appropriate public database (see <http://msb.embopress.org/authorguide#a3.5>) and the respective accession numbers should be listed in Materials & methods in a "Data availability" sub-section.

We appreciate that you provide the source code for DCQ on your website. For long-term archival purpose, we would be grateful if you could also provide it in the supplementary section as a zip archive (with a README file at the top level of the archive).

If you feel you can satisfactorily deal with these points and those listed by the referees, you may wish to submit a revised version of your manuscript. Please attach a covering letter giving details of

the way in which you have handled each of the points raised by the referees. A revised manuscript will be once again subject to review and you probably understand that we can give you no guarantee at this stage that the eventual outcome will be favorable.

 REFEREE REPORTS:

Reviewer #1:

The study introduces a new deconvolution method named Digital Cell Quantifier (DCQ). The method uses mRNA levels of immune cell surface marker genes and that of mixture samples as input to predict the composition of mixture samples. The method is novel in that it regularizes linear regression so reduces the number of parameters of the model. The authors first tested the performance of DCQ on synthetic mixtures of 6 cell types (5 immune cell types and non-immune cells). Then, the authors predicted the dynamics of 213 immune cell types of different origins during Influenza infection using 61 cell surface marker genes; the predictions recapitulated the documented immune cell dynamics in literature. Finally, the authors investigated the novel role of dendritic cells (DC) using Influenza infection; the flow cytometry-based investigation was initiated from the predicted dynamics of DC using DCQ.

The manuscript is well-written although some typos need to be corrected (see the minor comments). Critical references for previous work and methods are included in the manuscript. Figures align with the authors' conclusions.

The major comments for the manuscript are below:

1. The authors claimed that DCQ can discern closely related immune subtypes, but did not assess how closely related the cell types are. A correlation analysis between the gene expression profiles of input cell types would answer the question.
2. The authors compared the gene expression levels and documented presence and absence of proteins across the cell types of interest. The p-values are tabulated in Table S2. How many samples were used for this analysis? Can the authors plot the mRNA expression levels vs. protein expression to show the relation between mRNA and protein expression?
3. For Figure 2b, does "FACS measurements" mean input cell frequencies? Otherwise, how were the FACS measurements obtained? Why were DCQ predictions not compared to the input percentages for creating the mixed samples, but compared to FACS measures? Clarification is needed.
4. For Figure 2b, for one cell type, I assume that each data point represents on mixed sample. Why there are 10 data points for NK cells and CD8 T cells, but 8 data points for the other cell types? Can the authors tabulate the input percentages for each mixed sample?
5. For Figure 2b, why are there cell quantities in negative? Are those standardized values? Also, please change R in the figure to R² values to be consistent with the text (see page 5 line 22 of the manuscript).
6. To examine in vivo dynamics of Influenza pathogenesis, the authors used microarray data for the compendium cell types and RNAseq data for the mixture samples. How did the authors ensure that two datasets of different techniques could be combined?
7. Figure 5a: while DCQ predicted CD103-CD11b+ cDC, CD8+ pDC and CD8- pDC well, the prediction for CD103+CD11b- cDC captured experimental results poorly. What are the potential reasons for the poor prediction of CD103+CD11b- cDC? From Figure 5b, the two cDC populations seem to have similar gene expression profiles, is this one of the reasons?
8. Page 6: The authors discuss how the disease outcome phase is characterized by either elimination of viral load and decrease of disease symptoms or continued reduction in body weight and death.

Can DCQ predict disease prognosis based on early gene expression dynamics of the immune cell network?

9. Page 7: The authors found that many progenitor B cell subsets reduce during infection while the effector B cell subsets are increasing. Can DCQ be used to predict whether the progenitor numbers are decreasing as they are differentiating to effector subtypes without replenishment from the stem cell compartment due to the infectious stress on the hematopoietic system or that the effector cell compartment is solely proliferating?

10. Page 8: It is very interesting that DCQ can predict increased immune cell migration from different tissues into the lung compartment in duress. However, the authors detect reduced DP cell numbers in lung tissue, a highly unusual observation given that DPs are primarily found in the thymus and do not exit the organ without undergoing negative selection. Can the authors cite any existing literature showing DP cells are capable of circulation in the body?

11. Page 9: the authors found that the FACS-based subset of markers outperform alternative strategies for choosing input genes. What is the minimum number of cell surface markers needed to achieve desired prediction accuracy? How is the number of DCQ comparing to that for the regular linear regression model (e.g., Abbas et al.), or other recently published deconvolution models (Qiao et al., 2012)?

12. Page 11 line 5: the results for CCL22, ligands for CCR4 and chemo-attractants for B and T cells are not found in the manuscript? Please include the results in a reviewable format.

13. Page 11: The authors mention that CD103+CD11b- cDCs enter an exhaustion-like state at 3 days post infection. It would be helpful for the reader if the genes in Fig.5d could be grouped or color-coded according to anti-viral response, cytokine production, apoptosis or senescence to clearly demonstrate this point. Does the gene expression data indicate that the virus causes apoptosis of this DC cell type?

14. The details of the model algorithm should be included in the supplementary section of the manuscript, and the code should be provided for comparative purposes to other published algorithms.

The minor comments for the manuscript are below:

1. Page 2 line 20, "...but they fail when cell type cannot be easily decomposed...": change "decomposed" to "distinguished"
2. What is the relationship between Table S4 and Table S3? More description may be needed for the legend of Table S4.
3. Page 7 line 6: How was the "reshuffling test" performed?
4. Page 12 line 5 "...such as (i) Different ...": lower case D
5. Page 18 Figure 2 legend change "DC8+" to "CD8+"
6. The authors should consider including Massberg S. et al. Cell 2007 "Immunosurveillance by Hematopoietic Progenitor Cells Trafficking through Blood, Lymph, and Peripheral Tissues" in their references. The data in Fig.3 demonstrating the early temporal increase in the HSC compartment in the lung tissue is an excellent corroboration of the published results on HSC immunosurveillance, proliferation and differentiation to tissue-resident dendritic cells by Massberg et al.

Reviewer #3:

In this paper, Altboum and colleagues developed an approach to perform quantification of immune

cell population frequencies using gene expression data. The algorithm was first tuned and tested using artificial immune-cell mixtures. Next by using immune cell expression profiles from the ImmGen project (and another publication), the authors applied their approach to RNAseq data obtained from the whole lungs of influenza infected mice to study the infection dynamics post infection. They predicted some of the expected changes, but also new ones involving DC subsets. They then further evaluated these predictions as well as different approaches for choosing gene markers for the algorithm by comparison with cell types previously implicated in responses to infection.

The issue of deconvolving gene expression measurements of complex cell type mixtures into cell frequencies (and the related issue of deconvolving cell type specific expression profiles) is an important one. Various approaches have been proposed in the past, typically by starting from a set of well-defined cell-type specific gene expression profiles (e.g., Abbas et al. 2009 - also referred to by the authors). Here the authors are extending the basic linear-modeling approach in order to scale up the inference to hundreds of immune-cell subsets. However, while some performance data are included (e.g., Fig S2), some of the critical methodological details are missing, both in the manuscript and the methods. For example, the detailed formulation of the Lasso and Elastic Net models was not discussed. They mentioned that the different alpha and lambda showed "similar results and performance" but no concrete data was shown. The ranges of alpha and lambda are large (even if one omits alpha=1 and lambda=0 for LASSO fit). There is also no mention about what tool they used to perform this analysis. In terms of selecting optimal alpha and lambda, one could use cross-validation. It would also be helpful if the authors had a brief introduction of Elastic net for the readers. In addition, it was unclear how gene expression data from ImmGen were processed so that they can be integrated with RNAseq data to properly infer relative cell frequencies - how robust are the results when different types of normalizations were used (e.g., a x-fold change in RNAseq space may not be the same as a x-fold change in relative expression in the compendium)?

The motivation of using regularized regression was to avoid over-fitting, but the issue of over-fitting is quite specific in this context of using only ~60 FACS marker genes for inferring cell frequencies. For instance, if more genes were used as in previous approaches (e.g., those differentially expressed in at least one cell type), previous methods based on linear regression can readily be used because there will be more equations than unknowns. Similarly, in comparing their methods to others, it was based on using Lasso/Elastic net, which is likely more optimal when using a small number of genes as in this paper and therefore it would be difficult to draw conclusions about method comparisons here when only Lasso/Elastic net based approaches were used. Since multiple methods have been developed in the past, it's critical that the authors demonstrate both novelty and biological utility in their approach - e.g., how using FACS markers + Lasso compare to using differentially expressed genes + linear regression as in previous approaches (i.e., not simply fixing the modeling approach to Elastic Net and Lasso and change the genes).

Other comments:

In Fig 2/3, the actual scale and units of up/down in predicted cell freq. is not shown.

How were the protein expression of FACS markers across cell types established for computing the correlations shown in Figure S2? What were the directions of the correlations in addition to p values?

What were the coefficients/weights of the markers in the Elastic net models for different populations?

Concerning DCQ accuracy assessment using previously annotated cell types: why focusing on just the increasing cell types? What type of infection studies were included from the literature? Or was it just influenza? Manually selecting studies to include can lead to bias (e.g., the flow cytometry markers used, etc.) - how did the author assess and to avoid such potential biases? Again here the evaluation strategy for different methods cannot focus solely on using the FACS marker genes vs. other gene sets.

Overall, the DCQ predictions for the "known" increasing cell types isn't all that impressive. I would

like to see them divide the "remaining" cells into "unknown" and "known decrease/no change" sets, to get an idea of how DCQ is predicting the dynamics of subsets that don't change. Instead they focus only on the increasing. They do validate 4 of the populations in the non-known increasing set, but that is only 4 chosen out of many.

On page 8, top paragraph, the authors claimed that DCQ inferred an increase in the quantities of spleen and liver NKT cells but not thymic NKT cells. From figure 3 and supplemental figure 6d, it seems like the markers for gating thymic NKT subsets were different than those of spleen and liver (CD44 & NK1.1 Vs. CD4). The authors mentioned that their observation holds for all subsets of NKTs regardless of their levels of markers. Since some of the markers were different, it was not clear what this statement meant. Further data to clarify this would be helpful.

1st Revision - authors' response

22 January 2014

Dear Dr. Lemberger,

We are submitting the revised paper entitled “*Digital cell quantification identifies global immune cell dynamics during Influenza infection*” for your consideration. We were happy to see that the reviewers are enthusiastic about the technical advance of our work, its importance and impact. Following the thoughtful comments that were raised, we revised the text and performed additional analysis. We briefly summarize here the key questions raised by the reviewers and the revisions to the manuscript. A detailed point-to-point response follows below.

1. DCQ applicability for differential gene expression data. Along the manuscript and figures, we didn’t state clearly enough that DCQ is designed for *differential* gene expression data. In accordance, DCQ predicts *relative* cell quantities of each of the cell types in the analysis (rather than absolute quantities). We revised the text to emphasize this point more clearly.

2. Terminology refinement of DCQ scores. For clarification reasons, we refined our terminology for the two different scores provided by DCQ:

a. *Predicted relative cell quantities*, namely the change in the amount of each cell type before and after infection.

b. *Robustness score*, which stands for the significance of a predicted change in quantity (P-value score).

3. Selection of parameters. Reviewer #3 requested additional explanations regarding the DCQ algorithm and the elastic net model. We now added the details of the analysis to the revised manuscript and also highlight it in new Supplementary Figure 12. In particular, we defined two new objective functions that quantify a “good” parameter setting: (1) ‘*Quality of marker prediction*’ which is the standard objective function to fit the true value of the markers; (2) ‘*Quality of cell quantity prediction*’, namely fitting the predicted and true relative cell quantities (the predictors’ coefficients produced by the model). **New Supplementary Figure 12a.** illustrates that there is a clear tradeoff between these two objective functions and therefore we chose a set of parameters that would fit the balance between them. Furthermore, we tested the sensitivity of our method to different set of parameters. We show that when testing a large range of parameters,

over 75% attain better results than alternative methods (**new Supplementary Figure 12b**).

These results indicate that DCQ performance are not specific to particular combinations of parameters and can be attained using a range of parameter settings.

4. Integration of the cell compendium and transcription profiles of Influenza-infected mice. The reviewers asked about the integration of the different data sets used in our analysis. In fact, we have pre-processed each dataset independently and no additional data transformation was applied to combine them. In the revised manuscript, we added a new analysis to show that our method is robust to rescaling of each of these datasets. We tested both additive and multiplicative factors for rescaling both the differential RNA-seq data and the raw RNA-seq data. As shown in **new Supplementary Figure 13**, our results are stable when using different types of scaling.

5. Clarity of the text. All reviewers pointed out places in the manuscript that needed additional clarification. We apologize for that, and now address in full. **First**, we refined the terminology of different aspects in the algorithm as mentioned above throughout the manuscript. **Second**, we added a **new Supplementary Information no. 1**, which includes more details regarding the DCQ algorithm as well as the elastic net and lasso models. **Third**, we revised the Results and Methods sections to clarify the different approaches used to evaluate DCQ performance. **Finally**, we edited the text and figures to clarify all the sections indicated by the reviewers.

6. Data and source code. As requested, we uploaded the data in GEO (GSE49934) (**p. 20**) and now give the source code as **new Supplementary Information no. 2**.

Detailed responses to reviewers

Response to Reviewer #1:

The study introduces a new deconvolution method named Digital Cell Quantifier (DCQ). The method uses mRNA levels of immune cell surface marker genes and that of mixture samples as input to predict the composition of mixture samples. The method is novel in that it regularizes linear regression so reduces the number of parameters of the model.

The authors first tested the performance of DCQ on synthetic mixtures of 6 cell types (5 immune cell types and non-immune cells). Then, the authors predicted the dynamics of 213 immune cell types of different origins during Influenza infection using 61 cell surface marker genes; the predictions recapitulated the documented immune cell dynamics in literature. Finally, the authors investigated the novel role of dendritic cells (DC) using Influenza infection; the flow cytometry-based investigation was initiated from the predicted dynamics of DC using DCQ.

The manuscript is well-written although some typos need to be corrected (see the minor comments). Critical references for previous work and methods are included in the manuscript. Figures align with the authors' conclusions.

We thank the reviewer for this kind comment.

The major comments for the manuscript are below:

1. The authors claimed that DCQ can discern closely related immune subtypes, but did not assess how closely related the cell types are. A correlation analysis between the gene expression profiles of input cell types would answer the question.

The reviewer raised a valid concern regarding DCQ's ability to discern closely related cell types. We agree with the reviewer that it is important to evaluate this ability, and therefore we added a **new Supplementary Figure 8**. In **Supplementary Figure 8a**, we present a correlation matrix between the expression profiles of input cell types. **Supplementary Figure 8b,c,d** highlights three examples of closely related cell types (based on their gene expression profile), which show low similarity in their relative cell quantity (based on DCQ's predictions), demonstrating DCQ's ability to discern similar gene expression signatures. We have also revised the text accordingly (Results section, **p. 7-8**).

2. The authors compared the gene expression levels and documented presence and absence of proteins across the cell types of interest. The p-values are tabulated in Table

S2. How many samples were used for this analysis? Can the authors plot the mRNA expression levels vs. protein expression to show the relation between mRNA and protein expression?

We apologize for the lack of clarity on this subject; to the best of our knowledge, there is no published dataset containing measurements of the protein abundance for each of the immune cells included in our analysis. Therefore, we used the term ‘protein abundance’ to refer only to qualitatively known FACS intensities (as a binary state) of proteins that are known as cell surface markers of the various immune cell types (Murphy, 2012; Kindt *et al*, 2007; Benoist *et al*, 2012). As suggested, we added a **new supplementary Figure 3** containing a comparison between known FACS intensities of proteins (from literature) and expression level data of a few cell surface proteins. Each plot illustrates the difference in gene expression between immune cell types in which a specific cell surface protein is known to be present and the cell types in which it is likely absent. We evaluated this difference using t-test as detailed in **Supplementary Table 2**. We revised the text accordingly (**p. 4-5**).

3. For Figure 2b, does "FACS measurements" mean input cell frequencies? Otherwise, how were the FACS measurements obtained? Why were DCQ predictions not compared to the input percentages for creating the mixed samples, but compared to FACS measures? Clarification is needed.

The source of confusion was our lack of clarity in the description of **Figure 2**. We agree with the reviewer that we haven’t used the appropriate term for describing the input cell frequencies. As the reviewer mentioned, we mistakenly referred to the input cell frequencies as ‘FACS measurements’. In fact, DCQ’s predictions were indeed compared to the input relative cell quantity that we created the mixed samples with. We now carefully define this term and its calculation (**p. 5**, 2nd paragraph; and **p. 21** [Figure 2 legend]).

4. For Figure 2b, for one cell type, I assume that each data point represents on mixed sample. Why there are 10 data points for NK cells and CD8 T cells, but 8 data points for

the other cell types? Can the authors tabulate the input percentages for each mixed sample?

There are indeed ten data points for each of the five different cell types, but some of the points are located in a very similar position and hence the confusion. Following the reviewer's suggestion, we added a new **Supplementary Table 3** that contains the relative input cell quantity for each of the immune cells that were mixed in each of the ten experiments, and the relative quantity predicted by DCQ (see **p. 5** [manuscript] and Supplementary Tables legends **p. 23** [Supplementary Material]).

5. For Figure 2b, why are there cell quantities in negative? Are those standardized values? Also, please change R in the figure to R² values to be consistent with the text (see page 5 line 22 of the manuscript).

We apologize for lack of clarity regarding **Figure 2**. The DCQ method is applicable for *differential* gene expression data, and in accordance, predicts *relative* cell quantities. Therefore, DCQ predictions (presented in **Figure 2a, b** and in the **new Supplementary Table 3**) were obtained using differential gene expression, and were compared to relative input cell quantity. Results were generated as follows: **First**, we created a reference RNA-seq sample consisting of equal quantities for all input cell types. We used it to normalize all RNA-seq samples (the '*differential gene expression*' values). **Second**, DCQ was applied to this differential data and therefore its output represents relative cell quantity of the test sample compared to the reference sample (the '*predicted relative cell quantity*'). **Finally**, we calculate the '*input relative cell quantity*', namely the input cell quantity in the test sample compared to the reference sample. **Figure 2a,b** compared the input and predicted relative cell quantities.

We now corrected the terminology referring to the input relative cell quantities, and added a description of this calculation process in the main manuscript and in **Figure 2** (**p. 5**, 2nd paragraph; and **p. 21** [Figure 2 legend]).

Furthermore, following the reviewer's comment, we realized that it is not clear that DCQ's input data is differential (rather than absolute) gene expression data. We therefore now stress this in the Introduction Section (**p. 2**), Results section (**p. 4**, 1st paragraph; **p. 5**,

2nd paragraph; and **p. 6-7**), Methods (**p. 15-16**, paragraph of ‘The DCQ algorithm’) and Figure 1 legend (**p. 21**).

Regarding the change of the Pearson correlation coefficient to R-squared, this is a typo and should be R. We fixed this typo.

6. To examine in vivo dynamics of Influenza pathogenesis, the authors used microarray data for the compendium cell types and RNAseq data for the mixture samples. How did the authors ensure that two datasets of different techniques could be combined?

The reviewer has raised a valid concern here that relates to the integration of the gene expression data in the compendium of immune cell types and the differential gene expression data from the complex tissue. We have pre-processed each dataset independently, without any additional technique to combine them. However, our method is generally robust to rescaling of each of these datasets. To show this robustness, we added a new analysis aiming to check the effect of rescaling one of these datasets, as detailed below.

I. Rescaling the differential RNA-seq data. We tested an additive and multiplicative factors for rescaling the differential gene expression data. In each case, we evaluated the robustness of predictions in terms of the correlation between DCQ’s scores before and after scaling. Two DCQ scores were evaluated: (i) the predicted relative cell quantities, and (ii) the robustness (P value) score. In all four cases, we conclude that our results are stable when using different types of normalization of the RNA-seq data. We added a **new Supplementary Figure 13** presenting these results (a- multiplicative scaling, robustness scores, b – multiplicative scaling, predicted relative cell quantities c - additive scaling, robustness scores, d – additive scaling, predicted relative cell quantities), and briefly discuss it in the manuscript (**p. 19**).

II. Rescaling the original log RNA-seq data. Notably, such rescaling either has no effect or is equivalent to our experiments in the above case no. I. Rescaling with an additive constant (in log scale, before calculating differential expression) has no effect on the differential log values. Rescaling with a multiplicative factor is equivalent to

multiplicative factor applied on the differential expression data, as already demonstrated in **new Supplementary Figure 13a and b**.

We now clarify that no pre-processing was applied to integrate the two datasets (**Methods, p. 15**, paragraph of 'Pre-processing of RNA-seq data paragraph').

7. Figure 5a: while DCQ predicted CD103-CD11b+ cDC, CD8+ pDC and CD8- pDC well, the prediction for CD103+CD11b- cDC captured experimental results poorly. What are the potential reasons for the poor prediction of CD103+CD11b- cDC? From Figure 5b, the two cDC populations seem to have similar gene expression profiles, is this one of the reasons?

We agree that the predictions for CD103+CD11b- cDC is not as accurate as the other DC subtypes. We see one major potential reason for this lower accuracy, we find only a few hundred CD103+CD11b- cDC in the lungs both before and after infection. These low numbers may reduce the accuracy of FACS, DCQ or both. We slightly modified the caption in **Supplementary Table 6** to clarify this point (Supplementary Tables legends, **p. 24** [Supplementary Material]).

8. Page 6: The authors discuss how the disease outcome phase is characterized by either elimination of viral load and decrease of disease symptoms or continued reduction in body weight and death. Can DCQ predict disease prognosis based on early gene expression dynamics of the immune cell network?

The reviewer raises here a very relevant question, which we aim to follow in future research. Indeed a promising direction of research is classifying individuals based on early changes in cell quantities (rather than based on early changes in gene expression data or viral titer, *e.g.*, Ivan *et al*, 2012). We now add this important point into the discussion section (**p. 12**, 2nd paragraph).

9. Page 7: The authors found that many progenitor B cell subsets reduce during infection while the effector B cell subsets are increasing. Can DCQ be used to predict whether the progenitor numbers are decreasing as they are differentiating to effector subtypes

without replenishment from the stem cell compartment due to the infectious stress on the hematopoietic system or that the effector cell compartment is solely proliferating?

DCQ cannot discriminate between these two potential possibilities. To test for this, one would need to use DCQ predictions and validate these two options experimentally. We now clarify this point (Discussion, **p. 12-13**).

10. Page 8: It is very interesting that DCQ can predict increased immune cell migration from different tissues into the lung compartment in duress. However, the authors detect reduced DP cell numbers in lung tissue, a highly unusual observation given that DPs are primarily found in the thymus and do not exit the organ without undergoing negative selection. Can the authors cite any existing literature showing DP cells are capable of circulation in the body?

There are several high profile papers that state double positive lymphocytes cells have been characterized in the peripheral blood and secondary lymphoid tissues of several species such as human, mice, rat, macaque and swine during viral infections (Nascimbeni *et al*, 2004; Kenny *et al*, 2000; Bismarck *et al*, 2012; Zuckermann, 1999). However, their functionality is not yet determined. As stated in the manuscript, we think that our findings of the reduction in quantity of DP T cells can provide a potential mechanism by which peripheral progenitors collect information in the infected tissue before they differentiate or lead to activation of other specific cells. We added the citations mentioned to the manuscript and revised this part in the text (**p. 9**, 1st paragraph).

*11. Page 9: the authors found that the FACS-based subset of markers outperform alternative strategies for choosing input genes. What is the minimum number of cell surface markers needed to achieve desired prediction accuracy? How is the number of DCQ comparing to that for the regular linear regression model (e.g., Abbas *et al.*), or other recently published deconvolution models (Qiao *et al.*, 2012)?*

We thank the reviewer for this suggestion. We added a **new Supplementary Figure 10**, which illustrates the connection between the percent of markers used in each method and

the false- and true- positive rates achieved. Using 40% of the markers and above, DCQ algorithm successfully attains a much lower false positive rate and a higher true positive rate compared to other deconvolution method (Abbas *et al*, 2009; Lu *et al*, 2003). We have edited the text to include these results (**p. 10**, 2nd paragraph), added **new Supplementary Figure 10**, and also a clarification regarding the number of markers used in each of these methods (**p. 9-10**, ‘Evaluation of DCQ accuracy in vivo using previously annotated cell types’).

Notably, despite our attempts, we could not use Qiao *et al*, 2012 code in our setting, mainly due to an ‘out of memory’ problem when applying this method for a large number (hundreds) of cell types.

12. Page 11 line 5: the results for CCL22, ligands for CCR4 and chemo-attractants for B and T cells are not found in the manuscript? Please include the results in a reviewable format.

As suggested, we have now added a **new Supplementary Figure 11** containing the result for CCL22.

13. Page 11: The authors mention that CD103+CD11b- cDCs enter an exhaustion-like state at 3 days post infection. It would be helpful for the reader if the genes in Fig.5d could be grouped or color-coded according to anti-viral response, cytokine production, apoptosis or senescence to clearly demonstrate this point. Does the gene expression data indicate that the virus causes apoptosis of this DC cell type?

We now color code the genes according to the molecular function to improve visibility for the readers (**Figure 5d** and **p. 22-23** [Figure 5 legend]). We cannot conclude from this data that the DCs are undergoing apoptosis following infection.

14. The details of the model algorithm should be included in the supplementary section of the manuscript, and the code should be provided for comparative purposes to other published algorithms.

A detailed methodological description was indeed lacking in the original submission. Following the reviewer's suggestion, we added a **new Supplementary Information no. 1** containing the details of the DCQ algorithm. We now refer to this new supplement in **p. 15-16** ('The DCQ algorithm', 1st paragraph).

Regarding the source code of our method, we have provided DCQ as a source code and as a web-based software tool (DCQ.tau.ac.il). We also added a **new Supplementary Information no. 2** containing the source code. We also now updated the text in **p. 15-16** ('The DCQ algorithm', 1st paragraph) accordingly.

The minor comments for the manuscript are below:

15. Page 2 line 20, "...but they fail when cell type cannot be easily decomposed...": change "decomposed" to "distinguished"

We gladly accepted this comment.

16. What is the relationship between Table S4 and Table S3? More description may be needed for the legend of Table S4.

The confusion stems from the fact that DCQ provide two alternative scores, each table presents a different score. In detail, DCQ proceeds in two steps: **First**, calculating the regression (elastic net) coefficients, called '*predicted relative cell quantities*'. **Next**, calculating whether each coefficient significantly differs from zero (a $-\log$ P-value score). To avoid confusion, we now refer to the latter score with the term '*robustness score*'. We now updated the refined terminology throughout the manuscript (**p. 5, 7, and p. 16**).

Referring to **Supplementary Tables no. 3 and no. 4** (now **no. 4 and no. 5**, respectively), these tables refer to two distinct DCQ scores: the predicted relative cell quantities (**Supplementary Tables 4**) and robustness scores (**Supplementary Tables 5**). **Supplementary Table 4** contains all 213 cell types, whereas **Supplementary Table 5** contains only the significantly changing genes. We now refined the legends of these

Tables and the manuscript accordingly (see **p. 6-7** and **p. 16** [manuscript] and **Supplementary Tables 4, 5**).

17. Page 7 line 6: How was the "reshuffling test" performed?

We apologize for omitting this information. Permutation tests were performed by running DCQ on 10 permuted gene expression datasets and identifying significant cell types in each of these permuted datasets. The permuted datasets were generated by reshuffling the expression values of the cell surface markers among time points. This way we maintained the markers and the correlations among cell types in the compendium, while disrupting the correlation between markers and the order of time points. We have slightly revised the text in the **Results** section (**p. 7**) and added this explanation in the **Methods** section (**p.19**, 2nd paragraph).

18. Page 12 line 5 "...such as (i) Different ...": lower case D

As suggested, we have corrected this mistake.

19. Page 18 Figure 2 legend change "DC8+" to "CD8+"

The typo was corrected.

20. The authors should consider including Massberg S. et al. Cell 2007

"Immunosurveillance by Hematopoietic Progenitor Cells Trafficking through Blood, Lymph, and Peripheral Tissues" in their references. The data in Fig.3 demonstrating the early temporal increase in the HSC compartment in the lung tissue is an excellent corroboration of the published results on HSC immunosurveillance, proliferation and differentiation to tissue-resident dendritic cells by Massberg et al.

We greatly thank the reviewer for highlighting this important reference and now include it in the relevant section (**p. 9**).

Response to Reviewer #3:

In this paper, Altbourn and colleagues developed an approach to perform quantification of immune cell population frequencies using gene expression data. The algorithm was first tuned and tested using artificial immune-cell mixtures. Next by using immune cell expression profiles from the ImmGen project (and another publication), the authors applied their approach to RNAseq data obtained from the whole lungs of influenza infected mice to study the infection dynamics post infection. They predicted some of the expected changes, but also new ones involving DC subsets. They then further evaluated these predictions as well as different approaches for choosing gene markers for the algorithm by comparison with cell types previously implicated in responses to infection.

The issue of deconvolving gene expression measurements of complex cell type mixtures into cell frequencies (and the related issue of deconvolving cell type specific expression profiles) is an important one. Various approaches have been proposed in the past, typically by starting from a set of well-defined cell-type specific gene expression profiles (e.g., Abbas et al. 2009 - also referred to by the authors). Here the authors are extending the basic linear-modeling approach in order to scale up the inference to hundreds of immune-cell subsets.

1. However, while some performance data are included (e.g., Fig S2), some of the critical methodological details are missing, both in the manuscript and the methods. For example, the detailed formulation of the Lasso and Elastic Net models was not discussed.

We agree that this is an important point to refer to in our manuscript, and apologize for not providing it in the methodological description in the original submission. Following the reviewer's suggestion, we have added **new supplementary information no. 1** that clarifies the formulation of the lasso and elastic net models.

2. They mentioned that the different alpha and lambda showed "similar results and performance" but no concrete data was shown. The ranges of alpha and lambda are large (even if one omits alpha=1 and lambda=0 for LASSO fit). There is also no mention

about what tool they used to perform this analysis. In terms of selecting optimal alpha and lambda, one could use cross-validation.

The reviewer raises three valid concerns here, first, regarding the selection of the parameters for the model; second, regarding the sensitivity of results to the chosen parameters; and third, regarding the tools that were used to perform the analysis

1. Selection of model parameters. We agree with the reviewers that our original manuscript did not provide information or guidelines about selection of model parameters. In general, there are two possible objective functions for a “good” parameter setting. The standard objective function (commonly referred to as “cross validation” of the elastic net parameters) is to best fit the true value of the markers (the response vector). Here we were additionally interested in another objective function: fitting the predicted and true relative cell quantities (that is, the coefficients of the predictors produced by the model). We now added the definition of these two quality scores in the Methods section, referred to as ‘quality of a cell quantity prediction’ and ‘quality of a marker prediction’ (p. 19, 1st paragraph). We also added a **new Supplementary Figure 12a**, which shows that at least in our data, there is a clear tradeoff between these two objectives functions: A good-quality prediction of cell quantities can provide only a poor quality prediction of markers and vice versa. In this work, we chose a set of parameters that would hit the balance between the two objective functions, as indicated in the new **Supplementary Figure 12a**. We now clarify this and present **new Supplementary Figure 12a** in the **Methods** section (p. 19, 1st paragraph).

2. Sensitivity of results to the chosen set of elastic-net parameters. We agree with the reviewer that our original manuscript did not present the quality of DCQ when applied using many different sets of parameters. When testing a large number of parameter settings (all combination of lambda.min.ratio and alpha across the ranges 0.01-0.09 and 0.1-0.9), we find that using many combinations of parameter settings, our method outperforms the alternative methods on the complex lung tissue data (**see new supplementary Figure 12b**). For example, using 75% of the combinations of parameters, DCQ attains a better true positive rate compared to the alternative methods

(using FPR=0.1). These results indicate that DCQ performance are not specific to particular combinations of parameter settings and can be attained using a range of parameter settings. We now added a **new Supplementary Figure 12b** (referred in **p. 19**) that shows clearly the above results.

3. The tools that were used to perform the analysis. We apologize for omitting this important information. We have also revised the text to clarify this point (**p. 15-16**).

3. It would also be helpful if the authors had a brief introduction of Elastic net for the readers.

We apologize for not providing the necessary introduction in the methodological description in the original submission. As mentioned above, we have now added **new supplementary information no. 1**, which includes an introduction for elastic net.

4. In addition, it was unclear how gene expression data from ImmGen were processed so that they can be integrated with RNAseq data to properly infer relative cell frequencies - how robust are the results when different types of normalizations were used (e.g., a x-fold change in RNAseq space may not be the same as a x-fold change in relative expression in the compendium)?

The reviewer has raised a valid concern here that relates to the integration of the gene expression data in the compendium of immune cell types and the differential gene expression data from the complex tissue. We have pre-processed each dataset independently, without any additional technique to combine them. However, our method is generally robust to rescaling of each of these datasets. To show this robustness, we added a new analysis aiming to check the effect of transforming of one of the datasets on the outcome predicted results, as detailed below.

I. Rescaling the differential RNA-seq data. We tested an additive and multiplicative factors for rescaling the differential gene expression data. In each case, we evaluated the robustness of predictions in terms of the correlation between DCQ's scores before and after scaling. Two DCQ scores were evaluated: (i) the predicted relative cell quantities,

and (ii) the robustness (P value) score. In all four cases, we conclude that our results are stable when using different types of normalization of the RNA-seq data. We added a **new Supplementary Figure 13** presenting these results (a- multiplicative scaling, robustness scores, b – multiplicative scaling, predicted relative cell quantities c - additive scaling, robustness scores, d – additive scaling, predicted relative cell quantities), and briefly discuss it in the manuscript (**p. 19**).

II. Rescaling the original log RNA-seq data. Notably, such rescaling either has no effect or is equivalent to our experiments in the above case no. I. Rescaling with an additive constant (in log scale, before calculating differential expression) has no effect on the differential log values. Rescaling with a multiplicative factor is equivalent to multiplicative factor applied on the differential expression data, as already demonstrated in **new Supplementary Figure 13a and b**.

We now clarify that no pre-processing was applied to integrate the two datasets (**Methods, p. 15**, paragraph of ‘Pre-processing of RNA-seq data paragraph’).

5. The motivation of using regularized regression was to avoid over-fitting, but the issue of over-fitting is quite specific in this context of using only ~60 FACS marker genes for inferring cell frequencies. For instance, if more genes were used as in previous approaches (e.g., those differentially expressed in at least one cell type), previous methods based on linear regression can readily be used because there will be more equations than unknowns. Similarly, in comparing their methods to others, it was based on using Lasso/Elastic net, which is likely more optimal when using a small number of genes as in this paper and therefore it would be difficult to draw conclusions about method comparisons here when only Lasso/Elastic net based approaches were used. Since multiple methods have been developed in the past, it's critical that the authors demonstrate both novelty and biological utility in their approach - e.g., how using FACS markers + Lasso compare to using differentially expressed genes + linear regression as in previous approaches (i.e., not simply fixing the modeling approach to Elastic Net and Lasso and change the genes).

The main source of confusion regarding the comparison to alternative methods was our lack of clarity about the methods comparisons. For example, we did not compare DCQ to lasso with only one set of markers. Instead, we tested it with both the FACS-based markers and other larger marker sets, thus addressing the reviewer's suggestion. Similarly, we did not test linear regression with only the FACS-based markers. Instead, we tested linear regression with several alternative (small and large) sets of markers. Thus, our comparison indeed test several deconvolution formulations (lasso, elastic net, linear regression), each of which is tested using several alternative marker sets, including the FACS-based markers and the differentially expressed markers. Taken together, the current comparisons in the manuscript meet the reviewer's suggestion.

In detail, our comparisons include: (i) Evaluation of elastic net across five different strategies of marker selection (**p. 10, p. 18** and **p.22 [Figure 4b legend]**), (ii) Comparison of each of the prediction methods - elastic net, lasso and linear regression - across the five different strategies of marker selection (when possible) (**Supplementary Figure 9**), and (iii) Comparison to previous approaches, each approach was implemented using a different combination of prediction method and marker selection method: Abbas et al: non-regularized linear model with a group of 696 markers, Lu et al: non-regularized linear model with a group of 366 markers, and Nakaya et al: enrichment test with 20 markers per cell type (**Figure 4c**; see **p. 9-10, p. 18-19** and **p.22 [Figure 4c legend]**).

We realize now that the source of this confusion was our presentation of the different methods. We have now clarified the particular compared marker selection methods (new paragraph in the **Methods** section, **p. 18**, 'Influenza infection analysis' second paragraph) and also denote each one with a unique name ('All', 'Max', 'Max ratio' and 'random'), which is referred now in the **Results** section (**p. 9-10**) and also in **Figure 4b** and **Supplementary Figure 9**. We also revise the description of the compared methods and emphasize the combination of methods used in each comparison (**p. 9-10**, 'Evaluation of DCQ accuracy using previously annotated cell types' section). We have revised the legends of **Figure 4b,c** and **Supplementary Figure 9** accordingly.

Other comments:

6. *In Fig 2/3, the actual scale and units of up/down in predicted cell freq. is not shown.*

The scale is now added to the figures.

7. *How were the protein expression of FACs markers across cell types established for computing the correlations shown in Figure S2? What were the directions of the correlations in addition to p values?*

As we were not sure whether this question regards **Supplementary Figure 2** or **Supplementary Table 2**, we will provide answer for both options.

In **Supplementary Figure 2** we present the Pearson correlation (R and not R squared, direction is always positive) between the relative cell quantity predicted by each of the compared methods (elastic net, lasso and non-regularized linear regression) versus the input ('true') relative cell quantity. This comparison was performed using the collection of all 276 known cell surface markers, since non-regularized regression cannot be applied with a number of markers (observations) that is smaller than the number of cell types (predictors). We added this clarification to the manuscript (**p. 17**, 'In *silico* simulation' Methods section, 1st paragraph). This list of genes was obtained using: (i) The list of 61 genes that were used to isolate the cell types in the compendium (Benoist *et al*, 2012); and (ii) additional genes that characterize these immune cell types, based on two general literature references (Murphy, 2012; Kindt *et al*, 2007). We now clarify this point in the **Methods** section (**p. 17**, 'In *silico* simulation' Methods section, 1st paragraph).

Supplementary Table 2 presents the correspondence between the prior knowledge regarding the existence or absence of a specific marker on the cell surface of an immune cell types (based on known FACS intensities) and the pattern of its gene expression across all cell types. We evaluated this difference in gene expression using a t-test (the *P* values are shown in this table). All the *P* values shown are with respect to a positive correlation between existence and higher gene expression. We clarified the text to explain this point (**p. 4-5**, [manuscript] and **p.21-22**, [Supplementary Material]) and add a few examples in **new Supplementary Figure 3**.

8. *What were the coefficients/weights of the markers in the Elastic net models for different populations?*

DCQ is method for predicting the relative cell quantity of each of the immune cell types. Therefore the elastic net's coefficients are the '*predicted relative cell quantities*' rather than the markers (the markers are used here as the observations). We now added a **new Supplementary Information no. 1** that clarifies these details of the methodology. The coefficients of all the immune cell types included in our compendium are presented in **Supplementary Table 4**. We slightly modified the Results section (**p. 6-7**) to clarify this point.

9. *Concerning DCQ accuracy assessment using previously annotated cell types: why focusing on just the increasing cell types? What type of infection studies were included from the literature? Or was it just influenza? Manually selecting studies to include can lead to bias (e.g., the flow cytometry markers used, etc.) - how did the author assess and to avoid such potential biases?*

The reviewer raises two related concerns, regarding the assessment of DCQ accuracy using a possibly biased list of previously annotated cell types; and regarding the focus on only increasing cell types.

First, as the reviewer indicates, the set of increasing cell types must be constructed in an unbiased manner. We indeed constructed the increasing cell types set in a systematic unbiased manner. To avoid biases, we based our classification on extant knowledge from only two sources, thus ignoring the plethora of publications that specifically discuss one cell type or another. In particular, we use two criteria to classify a cell type into the increasing cells group: (i) cell types that were stimulated before profiling (45 cell types), (ii) cell types that are referred to as “Inflammatory” or “effectors” by either the Immgen consortium (Benoist *et al*, 2012) or by general two immunological reviews (Godfrey *et al*, 2010; Gordon & Taylor, 2005).

An equivalent systematic construction of “decreasing cells” is not feasible, as information about reduced cell quantity is scarce and do not appear in general reviews. Further, this

cannot be easily obtained from the “non-inflammatory” characteristic of a cell type. A list of decreasing cell types, therefore, cannot be systematically constructed.

We realized that the source of confusion is the lack of clarity regarding the systematic way we constructed the increasing cell list. We therefore added an explanation about the unbiased construction in the **Methods** section (**p. 18-19**) and indicate the source of information for each of the immune cell types (**Supplementary Table 1**).

10. Again here the evaluation strategy for different methods cannot focus solely on using the FACS marker genes vs. other gene sets.

The reviewer raises the same concern as in question 5. As mentioned in question 5, we do compare DCQ to several mathematical formulations, each of which is tested using several alternative methods for marker set selection (which produce different number of selected markers). We therefore fulfill the reviewer’s suggestion, as detailed in question 5.

11. Overall, the DCQ predictions for the "known" increasing cell types isn't all that impressive. I would like to see them divide the "remaining" cells into "unknown" and "known decrease/no change" sets, to get an idea of how DCQ is predicting the dynamics of subsets that don't change. Instead they focus only on the increasing. They do validate 4 of the populations in the non-known increasing set, but that is only 4 chosen out of many.

An equivalent systematic construction of “decreasing cells” is not feasible, as information about reduced cell quantity is scarce and do not appear in general reviews. Further, this cannot be easily obtained from the “non-inflammatory” characteristic of a cell type. A list of decreasing cell types, therefore, cannot be systematically constructed. We hope that DCQ now highlight such decreasing cell-types and they will be further experimentally validated in the future. We now clarify this point in the manuscript (**p. 18-19**)

12. On page 8, top paragraph, the authors claimed that DCQ inferred an increase in the quantities of spleen and liver NKT cells but not thymic NKT cells. From figure 3 and supplemental figure 6d, it seems like the markers for gating thymic NKT subsets were

different than those of spleen and liver (CD44 & NK1.1 Vs. CD4). The authors mentioned that their observation holds for all subsets of NKTs regardless of their levels of markers. Since some of the markers were different, it was not clear what this statement mean. Further data to clarify this would be helpful.

We agree with the reviewer and apologize for the lack of clarity. We agree that the different predicted cell quantities might be (i) a result of a real biological difference in quantities, or (ii) due to a related but different sub-population that was isolated with different markers in each of the tissues. In support of the former possibility, the two sub-populations are highly similar with respect to their expression profiles, and only differ in their predicted quantities (see **new Supplementary Figure 8d**). In the revised version we tone down our claims and clarify this point (**p. 8**, 1st paragraph).

References:

- Abbas AR, Wolslegel K, Seshasayee D, Modrusan Z & Clark HF (2009) Deconvolution of Blood Microarray Data Identifies Cellular Activation Patterns in Systemic Lupus Erythematosus. *PLoS One* **4**: 16
- Benoist C, Lanier L, Merad M & Mathis D (2012) Consortium biology in immunology: the perspective from the Immunological Genome Project. *Nat. Rev. Immunol.* **12**: 734–740
- Bismarck D, Schütze N, Moore P, Büttner M, Alber G & Buttlar H v. (2012) Canine CD4+CD8+ double positive T cells in peripheral blood have features of activated T cells. *Vet. Immunol. Immunopathol.* **149**: 157–166
- Godfrey DI, Stankovic S & Baxter AG (2010) Raising the NKT cell family. *Nat. Immunol.* **11**: 197–206
- Gordon S & Taylor PR (2005) Monocyte and macrophage heterogeneity. *Nat. Rev. Immunol.* **5**: 953–964
- Ivan FX, Rajapakse JC, Welsch RE, Rozen SG, Narasaraju T, Xiong GM, Engelward BP & Chow VT (2012) Differential pulmonary transcriptomic profiles in murine lungs infected with low and highly virulent influenza H3N2 viruses reveal dysregulation of TREM1 signaling, cytokines, and chemokines. *Funct. Integr. Genomics* **12**: 105–117
- Kenny E, Mason D, Pombo A & Ramírez F (2000) Phenotypic analysis of peripheral CD4+ CD8+ T cells in the rat. *Immunology* **101**: 178–184
- Kindt TJ, Goldsby RA & Osborne BA (2007) Kuby Immunology
- Lu P, Nakorchevskiy A & Marcotte EM (2003) Expression deconvolution: a reinterpretation of DNA microarray data reveals dynamic changes in cell populations. *Proc. Natl. Acad. Sci. U. S. A.* **100**: 10370–10375
- Murphy K (2012) Janeway's Immunobiology 8th edition. In *Garland Science* pp 1–75,127–156,387–428.
- Nascimbeni M, Shin E-C, Chiriboga L, Kleiner DE & Rehermann B (2004) Peripheral CD4(+)CD8(+) T cells are differentiated effector memory cells with antiviral functions. *Blood* **104**: 478–486
- Zuckermann FA (1999) Extrathymic CD4/CD8 double positive T cells. *Vet. Immunol. Immunopathol.* **72**: 55–66

2nd Editorial Decision

27 January 2014

Thank you again for submitting your work to Molecular Systems Biology. We are now satisfied with the modifications made and we will be pleased to accept your article for publication pending the following minor point:

- please include the accession number for the *RNA-Seq datasets* used in this study (cell type mixture and infection time-course). The text refers now only to "microarray data".

2nd Revision - authors' response

29 January 2014

Thank you for all your help with our manuscript. We have uploaded the revised manuscript and all other requested items to your website.